# DATA DEPENDENT RANDOMIZED SMOOTHING

## ABSTRACT

Randomized smoothing is a recent technique that achieves state-of-art performance in training certifiably robust deep neural networks. While the smoothing family of distributions is often connected to the choice of the norm used for certification, the parameters of these distributions are always set as global hyper parameters independent from the input data on which a network is certified. In this work, we revisit Gaussian randomized smoothing and show that the variance of the Gaussian distribution can be optimized at *each* input so as to maximize the certification radius for the construction of the smooth classifier. Since the data dependent classifier does not directly enjoy sound certification with existing approaches, we propose a memory-enhanced data dependent smooth classifier that is certifiable by construction. This new approach is generic, parameter-free, and easy to implement. In fact, we show that our data dependent framework can be seamlessly incorporated into 3 randomized smoothing approaches, leading to consistent improved certified accuracy. When this framework is used in the training routine of these approaches followed by a data dependent certification, we achieve 9% and 6% improvement over the certified accuracy of the strongest baseline for a radius of 0.5 on CIFAR10 and ImageNet.

## 1 INTRODUCTION

Despite the success of Deep Neural Networks (DNNs) in various learning tasks (Krizhevsky et al., 2012; Long et al., 2015), they were shown to be vulnerable to small carefully crafted adversarial perturbations (Goodfellow et al., 2015; Szegedy et al., 2013). For a DNN $f$ that correctly classifies an image $x$, $f$ can be fooled to produce an incorrect prediction for $x + \eta$ even when the adversary $\eta$ is so small that $x$ and $x + \eta$ are indistinguishable to the human eye. To circumvent this nuisance, there have been several works proposing heuristic training procedures to build networks that are *robust* against such perturbations (Cisse et al., 2017; Madry et al., 2018). However, many of these works provided a false sense of security as they were subsequently broken, *i.e.* shown to be ineffective against stronger adversaries (Athalye et al., 2018; Tramer et al., 2020; Uesato et al., 2018). This has inspired researchers to develop networks that are *certifiably robust*, *i.e.* networks that provably output constant predictions over a characterized region around every input. Among many certification methods, a probabilistic approach to certification called *randomized smoothing* has demonstrated impressive state-of-the-art certifiable robustness results (Cohen et al., 2019; Lecuyer et al., 2019; Li et al., 2019). In a nutshell, given an input $x$ and a base classifier $f$, *e.g.* a DNN, randomized smoothing constructs a "smooth classifier" $g(x) = \mathbb{E}_{\epsilon \sim \mathcal{D}} \left[ f(x + \epsilon) \right]$ such that, and under some choices of $\mathcal{D}$, $g(x) = g(x + \delta) \; \forall \delta \in \mathcal{R}$. As such, $g$ is certifiable within the certification region $\mathcal{R}$ characterized by $x$ and the smoothing distribution $\mathcal{D}$. While there has been considerable progress in devising a notion of "optimal" smoothing distribution $\mathcal{D}$ for when $\mathcal{R}$ is characterized by an $\ell_p$ certificate (Yang et al., 2020), a common trait among all works in the literature is that the choice of $\mathcal{D}$ is independent from the input $x$. For example, one of the earliest works on randomized smoothing grants $\ell_2$ certificates under $\mathcal{D} = \mathcal{N}(0, \sigma^2 I)$, where $\sigma$ is a free parameter that is constant for all $x$ (Cohen et al., 2019). That is to say, the classifier $f$ is smoothed to a classifier $g$ uniformly (same variance $\sigma^2$) over the entire input space of $x$. The choice of $\sigma$ used for certification is often set either arbitrarily or via cross validation to obtain best certification results (Salman et al., 2019a). We believe this is suboptimal and that $\sigma$ should vary with the input $x$ (data dependent), since using a fixed $\sigma$ may under-certify inputs (*i.e.* the constructed smooth classifier $g$ produces smaller certification radii), which are far from the decision boundaries as exemplified by $x_1$ in Figure 1. Moreover, this fixed $\sigma$ could be large for inputs

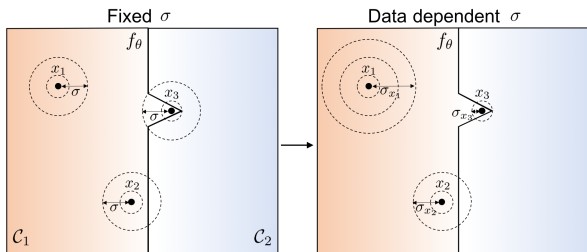

Figure 1: **From fixed to data dependent smoothing.** Using a fixed $\sigma$ to smooth $f_\theta$ for all inputs may under certify inputs (results in smaller certification radii), which are far from the decision boundary *e.g.* $x_1$, decrease prediction confidence *e.g.* $x_2$, or produce incorrect predictions *e.g.* $x_3$. Thus, smoothing should vary per input (right figure) to alleviate the aforementioned issues.

$x$ close to the decision boundaries resulting in a smooth classifier $g$ that incorrectly classifies $x$ (refer to $x_3$ in Figure 1).

In this paper, we aim to introduce more structure to the smoothing distribution $\mathcal{D}$ by rendering its parameters data dependent. That is to say, the base classifier $f$ is smoothed with a family of smoothing distributions to produce: $g(x) = \mathbb{E}_{\epsilon \sim \mathcal{N}(0, \sigma_x^2 I)} [f(x + \epsilon)]$ [1]. Note here that the variance of the Gaussian is now dependent on the data input $x$. Moreover, given that $\sigma_x$ varies with $x$, classical randomized smoothing based certification does not apply directly. We propose a simple memory-based approach to certify the resultant data dependent smooth classifier $g$. We show that our memory-enhanced data dependent smooth classifier can boost certification performance of several randomized smoothing techniques. Our contributions can thus be summarized in three folds. (**i**) We propose a parameter free and generic framework that can easily turn several randomized smoothing techniques into their data dependent variants. In particular, given a network $f$ and an input $x$, we propose to optimize the smoothing distribution parameters for every $x$, *e.g.* $\sigma_x^*$, so they maximize the certification radius. This choice of $\sigma_x^*$ is then used to smooth $f$ at $x$ and construct a smoothed classifier $g$. Moreover, as the data dependent smooth classifier is not directly certifiable using Cohen et al. (2019) MCMC approaches, we propose a memory-enhanced data dependent smooth classifier for certification. (**ii**) We demonstrate the effectiveness of our memory-enhanced data dependent smoothing by showing that we can improve the certified accuracy of several models, specifically models trained with Gaussian augmentation (COHEN) (Cohen et al., 2019), adversaries on the smoothed classifier (SMOOTHADV) (Salman et al., 2019a), and radius regularization (MACER) (Zhai et al., 2020) *without any model retraining*. We boost the certified accuracy of the best baseline by 5.4% on CIFAR10 and by 2.8% on ImageNet for $\ell_2$ perturbations with less than 0.5 (=127/255) ball radius. (**iii**) We show that incorporating the proposed data dependent smoothing in the training pipeline of COHEN, SMOOTHADV and MACER can further boost results to get certified accuracies of 68.3% on CIFAR10 and 64.2% on ImageNet at $\ell_2$ perturbations less than 0.25.

## 2 RELATED WORK

**Certified Defenses.** Certified defenses aim to guarantee that an adversary does not exist in a certain region around a given input. Certified defenses can be divided into exact (Cheng et al., 2017; Lomuscio & Maganti, 2017; Huang et al., 2017; Ehlers, 2017) and relaxed certification (Salman et al., 2019b; Wong & Kolter, 2018). Generally, exact certification suffers from poor scalability with networks that are at most 3 hidden layers deep (Tjeng et al., 2019). On the other hand, relaxed methods resolve this issue by aiming at finding an upper bound to the worst adversarial loss over all possible bounded perturbations around a given input (Weng et al., 2018). However, the latter is too expensive for any mixed certification-training routine.

**Randomized Smoothing.** The earliest work on randomized smoothing (Lecuyer et al., 2019) was from a differential privacy perspective, where it was demonstrated that adding Laplacian noise enjoys an $\ell_1$ certification radius in which the average classifier prediction under this noise is constant. This work was later followed by the tight $\ell_2$ certificate radius for Gaussian smoothing (Cohen et al., 2019). Since then, there has been a body of work on randomized smoothing with empirical defenses (Salman et al., 2019a) to certify black box classifiers (Salman et al., 2020). Other works derived certification

---

[1]The paper focuses on Gaussian smoothing, but the idea holds for other parameterized distributions.

guarantees for $\ell_1$ bounded (Teng et al., 2019), $\ell_\infty$ bounded (Zhang et al., 2019), and $\ell_0$ bounded (Levine & Feizi, 2020) perturbations. Even more recently, a novel framework that finds the optimal smoothing distribution for a given $\ell_p$ norm (Yang et al., 2020) was proposed showing state-of-art certification results on $\ell_1$ perturbations. We deviate from the common literature by introducing the notion of smoothing, particularly Gaussian smoothing for $\ell_2$ perturbations, which varies depending on the input. In particular, since an input $x$ that is far from the decision boundaries should tolerate larger smoothing (and equivalently have a larger certification radius) as compared to inputs closer to these boundaries, we optimize for the amount of smoothing per input (specifically $\sigma_x$) that maximizes the certification radius. This proposed process is denoted as *data dependent smoothing* where we provide a procedure for certifying the resultant smooth classifier.

## 3 DATA DEPENDENT SMOOTHING

### 3.1 PRELIMINARIES AND NOTATIONS

We consider the standard classification problem, where $x \in \mathbb{R}^d$ and the labels $y \in \mathcal{Y} = \{1, \ldots, k\}$ form the input-label pairs $(x, y)$ sampled from an unknown data distribution. Unless explicitly mentioned, we consider a classifier $f_\theta : \mathbb{R}^d \to \mathcal{P}(\mathcal{Y})$ parameterized by $\theta$ where $\mathcal{P}(\mathcal{Y})$ is a probability simplex over $k$ labels. We say that $f_\theta$ is $\ell_p^r$ certifiably accurate for an input $x$, if and only if, $\arg\max_c f_\theta^c(x) = \arg\max_c f_\theta^c(x+\delta) = y \ \forall \ \|\delta\|_p \le r$, where $f_\theta^c$ is the $c^{\text{th}}$ element of $f_\theta$. That is to say, the classifier correctly predicts the label of $x$ and enjoys a constant prediction for all perturbations $\delta$ that are in the $\ell_p$ ball of radius $r$ from $x$. As such, the overall $\ell_p^r$ certification accuracy is defined as the average certified accuracy over the data distribution. In this paper and following previous works (Cohen et al., 2019; Salman et al., 2019a; Zhai et al., 2020), we focus on $\ell_2^r$ certification.

### 3.2 OVERVIEW OF RANDOMIZED SMOOTHING

Randomized smoothing constructs a certifiable classifier $g_\theta$ by smoothing a base classifier $f_\theta$. That is to say, for any $\sigma > 0$, the smooth classifier is defined as follows: $g_\theta(x) = \mathbb{E}_{\epsilon \sim \mathcal{N}(0, \sigma^2 I)} [f_\theta(x+\epsilon)]$. Let $g_\theta$ predict label $c_A$ for input $x$ with some confidence, *i.e.* $\mathbb{E}_\epsilon[f_\theta^{c_A}(x+\epsilon)] = p_A \ge p_B = \max_{c \ne c_A} \mathbb{E}_\epsilon[f_\theta^c(x+\epsilon)]$, then, $g_\theta$ is certifiably robust at $x$ with certification radius:

$$R = \frac{\sigma}{2}\left(\Phi^{-1}(p_A) - \Phi^{-1}(p_B)\right). \tag{1}$$

Here, $g(x+\delta) = g(x) \ \forall \|\delta\|_2 \le R$, where $\Phi$ is the CDF of the standard Gaussian.

### 3.3 ROBUSTNESS-ACCURACY TRADE-OFF

Note that Equation 1 holds regardless of the prediction $c_A$ made by the smooth classifier $g_\theta$. This suggests that one can perhaps improve the robustness of $g_\theta$, *i.e.* increase certification radius $R$ where $g_\theta$ is constant, by increasing the hyper parameter $\sigma$ in Equation 1. However, to reason about $\ell_2^r$ certification accuracy, it is not enough to increase the certification radius $R$, as this requires $c_A$ to be the correct prediction for $x$ by $g_\theta$. This reveals the robustness-accuracy trade-off as one cannot improve $\ell_2^r$ certified accuracy by only increasing the certification radius $R$ (robustness) through the increase in $\sigma$. This is because it comes at the expense of requiring a classifier $g_\theta$ that correctly classifies $x$ with correct label $y$ under large Gaussian perturbations (accuracy). As such, the following inequality should hold $\mathbb{E}_\epsilon[f_\theta^y(x+\epsilon)] \ge p_A \ge p_B \ge \max_{c \ne y} \mathbb{E}_\epsilon[f^c(x+\epsilon)]$.

### 3.4 DATA DEPENDENT SMOOTHING FOR CERTIFICATION

The certification region $\mathcal{R} = \{\delta : \|\delta\|_2 \le R\}$ at an input $x$ is fully characterized by the classifier $f_\theta$ and the standard deviation of the Gaussian distribution $\sigma$. Moreover, for a given $f_\theta$, the certification region $\mathcal{R}$ varies at different $x$, when $\sigma$ is fixed, due to the nonlinear dependence of the prediction gap $\Phi^{-1}(p_A(x;\sigma)) - \Phi^{-1}(p_B(x;\sigma))$ on $x$. This hints that, for a given $f_\theta$, different inputs $x$ may enjoy a different optimal $\sigma_x^*$ that maximizes the certification region through radius $R$. To see this, consider the three inputs $x_1$, $x_2$ and $x_3$ all correctly classified by the binary classifier $f_\theta$ as $\mathcal{C}_1$ in Figure 1. Using a fixed $\sigma$ to smooth the predictions of $f_\theta$, *i.e.* predict with $g_\theta$, reveals that inputs, depending on how close they are from the decision boundaries, can enjoy different levels of smoothing without affecting the prediction of $g_\theta$. For instance, as shown in Figure 1 for constant $\sigma$, the input far from the decision

boundary $x_1$ could have still been classified correctly with similarly large prediction gap even if $f_\theta$ were to be smoothed with a larger $\sigma$. This indicates that perhaps the certification radius at $x_1$ could have been enlarged with a larger smoothing $\sigma$. As for $x_2$, we can observe that while the prediction under this choice of $\sigma$ by $g_\theta$ is still correct, the prediction gap $\Phi^{-1}(p_A(x; \sigma)) - \Phi^{-1}(p_B(x; \sigma))$ drops, due to having more Gaussian samples fall in the $\mathcal{C}_2$ region. This indicates that a different choice of $\sigma$ could have been used to trade-off the drop in prediction gap and certification radius. Last, for the input $x_3$ that is very close to the decision boundary, the sub optimal choice of $\sigma$ (too large for $x_3$) could result in an incorrect prediction by $g_\theta$. Despite the observations that $\sigma$ plays a significant role in $\ell_2^r$ certification accuracy, certification methods generally (**i**) choose $\sigma$ arbitrarily and (**ii**) set it to be constant for all $x$. Based on this observation, for a given smooth classifier with a specific $\sigma_0$, where $\sigma_0$ can be zero reducing the smooth classifier to $f_\theta$, we seek to construct another smooth classifier with parameter $\sigma_x^*$ for every input $x$ such that: (**i**) the prediction of both smooth classifiers (smoothing with $\sigma_0$ and $\sigma_x^*$) is identical for all $x$. (**ii**) The certification radius of the new smooth classifier at every $x$ is maximized. To construct a classifier smoothed with $\sigma_x^*$ enjoying the two previous

---

**Algorithm 1:** Data Dependent Certification

**Function** OptimizeSigma ($f_\theta$, $x$, $\alpha$, $\sigma_0$, $n$):

 **Initialize:** $\sigma_x^0 \leftarrow \sigma_0$, $K$

 **for** $k = 0 \ldots K - 1$ **do**

  sample $\hat\epsilon_1, \ldots \hat\epsilon_n \sim \mathcal{N}(0, I)$

  $\psi(\sigma_x^k) = \frac{1}{n} \sum_{i=1}^n f_\theta(x + \sigma_x^k \hat\epsilon_i)$

  $E_A(\sigma_x^k) = \max_c \psi^c; \; y_A = \arg\max_c \psi^c;$

  $E_B(\sigma_x^k) = \max_{c \neq y_A} \psi^c$

  $R(\sigma_x^k) = \frac{\sigma_x^k}{2} \left( \Phi^{-1}(E_A) - \Phi^{-1}(E_B) \right)$

  $\sigma_x^{k+1} \leftarrow \sigma_x^k + \alpha \nabla_{\sigma_x^k} R(\sigma_x^k)$

 $\sigma_x^* \leftarrow \sigma_x^K$

 **return** $\sigma_x^*$

---

properties, let $c_A$ be the prediction under $\sigma_0$ smoothing, *i.e.* $c_A = \arg\max_c \mathbb{E}_{\epsilon \sim \mathcal{N}(0, \sigma_0 I)}[f^c(x + \epsilon)]$. We directly maximize the radius $R$ in Equation 1 over $\sigma$ for every $x$ by solving the following optimization:

$$\sigma_x^* = \arg\max_\sigma \frac{\sigma}{2} \left( \Phi^{-1} \left( \mathbb{E}_{\epsilon \sim \mathcal{N}(0, \sigma^2 I)}[f_\theta^{c_A}(x + \epsilon)] \right) - \Phi^{-1} \left( \max_{c \neq c_A} \mathbb{E}_{\epsilon \sim \mathcal{N}(0, \sigma^2 I)}[f_\theta^c(x + \epsilon)] \right) \right). \tag{2}$$

Since $\Phi^{-1}$ is a strictly increasing function, it is important to note that solving Equation 2 for a fixed $c_A$ can at worst yield a smooth classifier of an identical radius to when the classifier is smoothed with $\sigma_0$ with both classifiers predicting $c_A$ for $x$.

**Solver**. While our proposed Objective 2 has a similar form to the MACER regularizer (Zhai et al., 2020) used during training, ours differs in that we optimize $\sigma$ for every $x$ and not the network parameters $\theta$, which are fixed here. A natural solver for 2 is stochastic gradient ascent with the expectation approximated with $n$ Monte Carlo samples. As such, the gradient of the objective at the $k^{\text{th}}$ iteration will be approximated as follows: $\nabla_{\sigma^k} \frac{\sigma^k}{2} \left[ \Phi^{-1} \left( \gamma^{c_A}(\sigma^k) \right) - \Phi^{-1} \left( \max_{c \neq c_A} \gamma^c(\sigma^k) \right) \right]$, where $\gamma^c(\sigma^k) = \frac{1}{n} \sum_{i=1}^n f^c(x + \epsilon_i)$ for $\epsilon_1, \ldots, \epsilon_n \sim \mathcal{N}(0, (\sigma^k)^2 I)$. However, this estimation of the gradient suffers from high variance due to the dependence of the expectation on the optimization variable $\sigma$ that parameterizes the smoothing distribution $\mathcal{N}(0, \sigma^2 I)$ (Williams, 1992). To alleviate this, we use the *reparameterization trick* suggested by Kingma & Welling (2014); Rezende et al. (2014) to compute a lower variance gradient estimate for our Objective 2. In particular, with the change of variable $\epsilon = \sigma \hat\epsilon$ where $\hat\epsilon \sim \mathcal{N}(0, I)$, Objective 2 is now *equivalent to*:

$$\sigma_x^* = \arg\max_\sigma \frac{\sigma}{2} \left( \Phi^{-1} \left( \mathbb{E}_{\hat\epsilon \sim \mathcal{N}(0, I)}[f_\theta^{c_A}(x + \sigma\hat\epsilon)] \right) - \Phi^{-1} \left( \max_{c \neq c_A} \mathbb{E}_{\hat\epsilon \sim \mathcal{N}(0, I)}[f_\theta^c(x + \sigma\hat\epsilon)] \right) \right) \tag{3}$$

Note that, unlike before, the expectation over the distribution $\hat\epsilon \sim \mathcal{N}(0, I)$ no longer depends on the optimization variable $\sigma$. This allows the gradient of 3 to enjoy a lower variance compared to the gradient of 2 (Kingma & Welling, 2014; Rezende et al., 2014). Algorithm 1 summarizes the updates for optimizing $\sigma$ for each $x$ by solving 3 with $K$ steps of stochastic gradient ascent. It is worthwhile to mention that the function OptimizeSigma in Algorithm 1 is agnostic of the choice of architecture $f_\theta$ and of the training procedure that constructed $f_\theta$.

### 3.5   Memory-Based Certification for Data Dependent Classifiers

Unlike previous approaches where $\sigma$ is constant for all inputs, the data dependent classifier $g_\theta$ with varying $\sigma$ per input can not be directly certified by the classical Monte Carlo algorithms

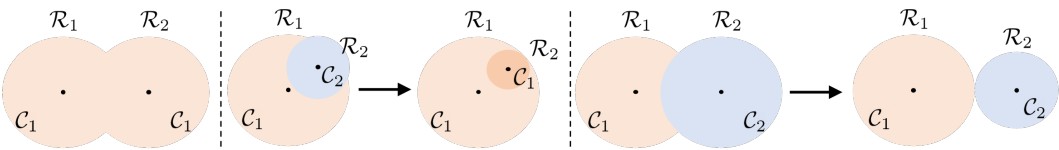

Figure 2: **Memory-based certification of the data dependent classifier.** Given a memory of an input $x_1$ with a certified region $\mathcal{R}_1$ and another input $x_2$ with a certified region $\mathcal{R}_2$. Three scenarios could arise where $\mathcal{R}_1$ and $\mathcal{R}_2$ intersect. **Left**: The certified regions intersect while both $x_1$ and $x_2$ share the same prediction. In this case, $x_2$ along with its certified region are directly added to memory. **Middle**: $x_2$ lies inside $\mathcal{R}_1$ with a different prediction from $x_1$. In this case, $x_2$ is predicted with the same prediction as $x_1$ and added to memory along with the largest subset of $\mathcal{R}_2$ that is within $\mathcal{R}_1$. **Right**: $x_2$ lies outside the $\mathcal{R}_1$ with a different prediction from $x_1$ In this case, $x_2$ with its prediction are added to memory along with the largest certified region in $\mathcal{R}_2$ not intersecting with $\mathcal{R}_1$.

proposed by Cohen et al. (2019). This is since the data dependent classifier $g_\theta$ does not enjoy a constant $\sigma$ within the given certification region, *i.e.* $g_\theta$ tailors a new $\sigma_x$ for every input $x$ including within the certified region of $x$. Informally, let $R(\sigma_{x_1}^*)$ be the radius of certification at $x_1$ granted by the data dependent classifier $g_\theta$. The data dependent classifier *does not guarantee* that there *can not exist* $x_2$ within the region of certification of $x_1$, *i.e.* $\|x_1 - x_2\|_2 \leq R(\sigma_{x_1}^*)$, where $g_\theta$ with $\sigma_{x_2}^*$ predicts $x_2$ differently from $x_1$ breaking the soundness of certification.

To circumvent this problem, we propose a memory-based procedure to certifying our proposed data dependent classifier. Let $\{x_i\}_{i=1}^N$ be a set of previously predicted inputs and $\{\mathcal{C}_i\}_{i=1}^N$ be their corresponding predictions with mutually exclusive $\ell_2$ certified regions $\mathcal{R}_i$ for differently predicted inputs, *i.e.* $\mathcal{R}_i \cap \mathcal{R}_j = \emptyset \ \forall i \neq$

---

**Algorithm 2:** Training with Data Dependent $\sigma_{x_i}$

---

**Function** `TrainBatch`$(f_\theta, \{x_i, y_i\}_{i=1}^B, \{\sigma_{x_i}\}_{i=1}^B, \alpha, n)$:

    **for** $i = 1, \dots, B$ **do**

        $\sigma_{x_i}^* = $ `OptimizeSigma`$(f_\theta, x_i, \alpha, \sigma_{x_i}, n)$

    `TrainFunction`$\left(\{x_i, y_i\}_{i=1}^B, \{\sigma_{x_i}^*\}_{i=1}^B\right)$ // any
        training routine e.g. MACER

---

$j, \mathcal{C}_i \neq \mathcal{C}_j$. Let $x_{N+1}$ be a new input with a certified region $\mathcal{R}_{N+1}$ computed by the Monte Carlo algorithms of Cohen et al. (2019) for the data dependent classifier $g_\theta$ with prediction $\mathcal{C}_{N+1}$. If there exists an $i$ such that $\mathcal{R}_{N+1} \cap \mathcal{R}_i \neq \emptyset$, $x_{N+1} \in \mathcal{R}_i$, and $\mathcal{C}_{N+1} \neq \mathcal{C}_i$, we adjust the prediction of the data dependent classifier $g_\theta$ to be $\mathcal{C}_i$ and update $\mathcal{R}_{N+1}$ to be the largest subset of $\mathcal{R}_{N+1}$ that is a subset of $\mathcal{R}_i$ (see middle example in Figure 2). On the other hand, if $\mathcal{R}_{N+1} \cap \mathcal{R}_i \neq \emptyset$, $x_{N+1} \notin \mathcal{R}_i$, and that $\mathcal{C}_{N+1} \neq \mathcal{C}_i$, we update $\mathcal{R}_{N+1}$ to be the largest subset of $\mathcal{R}_{N+1}$ not intersecting with $\mathcal{R}_i$ (see right example in Figure 2). We perform the previous operations for all elements in the memory and add $x_{N+1}, \mathcal{C}_{N+1}, \mathcal{R}_{N+1}$ to memory. The aforementioned procedure grants a sound certification for the data dependent classifier preventing by construction overlapping certified regions with different predictions.While the memory-based certification is essential for a sound certification, empirically, we never found in any of the later experiments a case where two inputs predicted differently suffer from intersecting certified regions. That is to say while our sound certificate works on the memory-enhanced data dependent smooth classifier, we found that the certified radius of the memory classifier for every input is the radius granted by the Monte Carlo certificates of Cohen et al. (2019) for the data dependent classifier. Therefore and throughout, we refer to the memory-enhanced data dependent smooth classifier and data dependent smooth classifier interchangeably. We elaborate more on this and provide an algorithm in the **Appendix**.

### 3.6 TRAINING WITH DATA DEPENDENT SMOOTHING

Models that enjoy a large $\ell_2^r$ certification accuracy under the randomized smoothing framework need to enjoy a large certification radius $R$ in Equation 1 for all $x$ and be able to correctly classify inputs corrupted with Gaussian noise, *i.e.* $g_\theta(x) = y$. While there are several approaches to train $f_\theta$ (or directly $g_\theta$) so as to output correct predictions for inputs corrupted with noise sampled from $\mathcal{N}(0, \sigma^2 I)$, all existing works fix $\sigma$ for all inputs during training. We are interested in complementing these approaches with smoothing distributions that are data dependent. As such, we can employ the training procedure of these approaches but with $\sigma_x^*$ computed by `OptimizeSigma`. Algorithm 2 summarizes this proposed training pipeline. The function `TrainFunction` proceeds by performing backpropagation using any training scheme, given the estimated $\sigma_{x_i}^*$ for each $x_i$. We note that whenever Algorithm 2 is used, we initialize $\sigma_{x_i}$ at each epoch with $\sigma_{x_i}^*$ computed at the previous

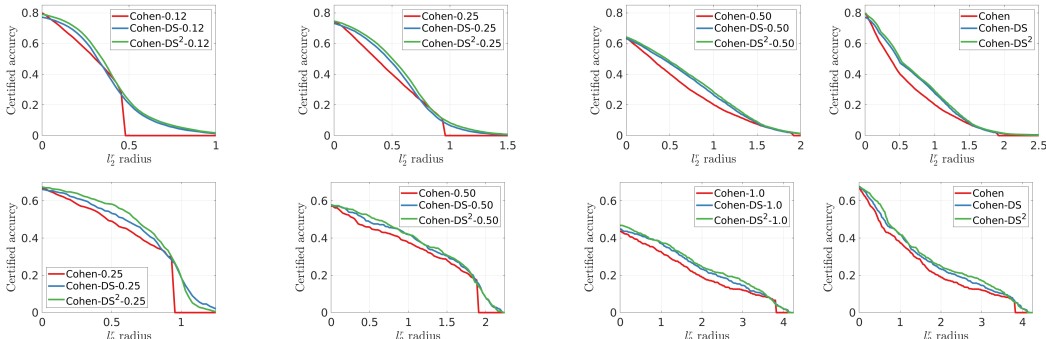

Figure 3: **Certified accuracy comparison against COHEN.** For several $\sigma$ values used in training (shown in the legend), we compare COHEN against our two variants: COHEN-DS with data dependent certification only and COHEN-DS$^2$, where data dependency is incorporated in both training and certification. We show CIFAR10 and ImageNet results in the first and second rows, respectively. The last column shows the envelope across $\sigma$.

epoch. Since COHEN, SMOOTHADV and MACER are among the most popular approaches that embed randomized smoothing certificates as part of the training routine, `TrainFunction` refers here to any of these three training methods. Empirically, we show that we can boost all three methods even further when models are trained with Algorithm 2.

## 4 EXPERIMENTS

We conduct two sets of experiments to validate our key contributions. (**i**) We show that we can boost certified accuracy for several pre-trained models by using Algorithm 1 for data dependent smoothing only during certification, *i.e.* without employing any additional training. (**ii**) Once data dependent smoothing is employed during training, we can improve the certified accuracy even further. Since our framework is agnostic to the training routine, we incorporate it into (**i**) COHEN (Cohen et al., 2019), (**ii**) SMOOTHADV (Salman et al., 2019a) and (**iii**) MACER (Zhai et al., 2020). Throughout, we use DS to refer to when data dependent smoothing is used only in certification and DS$^2$ when it is used during both training and certification.

**Setup.** We conduct experiments with ResNet-18 and ReNet-50 (He et al., 2016) on CIFAR10 (Krizhevsky & Hinton, 2009) and ImageNet (Russakovsky et al., 2015), respectively. For CIFAR10 experiments, we train from scratch for 200 epochs. For ImageNet, we initialize using the network parameters provided by the authors. When $\sigma$ is fixed and following prior art, *e.g.* COHEN, SMOOTHADV, and MACER, we set $\sigma \in \{0.12, 0.25, 0.50\}$ and $\sigma \in \{0.25, 0.50, 1.0\}$ for CIFAR10 and ImageNet, respectively, for training and certification. We set $\alpha = 10^{-4}$ in Algorithm 1 and the initial $\sigma_0$ to the $\sigma$ used in training the respective model. Unless stated otherwise, we set $n = 1$ in Algorithm 1. Following COHEN and SMOOTHADV, we compare models using the approximate certified accuracy curve (simply referred to as certified accuracy) followed by the envelope curve over all $\sigma$. We also report the Average Certified Radius (ACR) proposed by MACER $^1/_{|\mathcal{S}_{test}|} \sum_{(x,y) \in \mathcal{S}_{test}} R(f_\theta, x) . \mathbb{1}\{\arg\max_c g_\theta^c(x) = y\}$, where $\mathbb{1}\{.\}$ is an indicator function. Following COHEN[2] and all randomized smoothing methods, we certify all results using $N_0 = 100$ Monte Carlo samples for prediction and $N = 100,000$ estimation samples to estimate the radius with a failure probability of $0.001$ given a smoothing $\sigma$.

### 4.1 COHEN + DS

We combine data dependent smoothing with COHEN. Following Gaussian augmentation, this method trains $f_\theta$ on $(x + \epsilon)$, where $\epsilon \sim \mathcal{N}(0, \sigma^2 I)$, with the cross entropy loss.

**DS for certification only.** We first certify the trained models with the same fixed $\sigma$ used in training for all inputs, dubbed COHEN. Then, we certify using the memory based certification the same trained models with the proposed data dependent $\sigma_x^*$ produced by Algorithm 1, which we refer to as COHEN-DS. Figure 3 plots the certified accuracy for CIFAR10 and ImageNet in the first and second rows, respectively. Even though the base classifier $f_\theta$ is identical for COHEN and COHEN-DS, Figure

---

[2]We use the available code of Cohen et al. (2019) to report all certification results in the paper for a given $\sigma$.

Table 1: **Best certified accuracy per radius and ACR** of COHEN, COHEN-DS and COHEN-DS$^2$. FS denotes a fixed $\sigma$ in smoothing. DS denotes the use of a data-dependent $\sigma$.

| CIFAR10 | Radius Train | Radius Certify | 0.0 | 0.25 | 0.50 | 0.75 | 1.00 | 1.25 | 1.50 | 1.75 | 2.00 | 2.25 | ACR |
|---|---|---|---|---|---|---|---|---|---|---|---|---|---|
| COHEN | FS | FS | **79.9** | 58.3 | 40.1 | 29.2 | 20.2 | 13.1 | 7.3 | 3.3 | 0.0 | 0.0 | 0.591 |
| COHEN-DS | FS | DS | 77.2 | 64.5 | 47.8 | 38.3 | 27.6 | 16.5 | 8.0 | 3.2 | 1.2 | **0.7** | **0.784** |
| COHEN-DS$^2$ | DS | DS | 79.8 | **66.5** | **50.4** | **39.2** | **29.1** | **18.3** | **8.8** | **3.8** | **1.4** | 0.6 | 0.764 |

| ImageNet | Radius Train | Radius Certify | 0.0 | 0.25 | 0.50 | 0.75 | 1.00 | 1.50 | 2.0 | 2.5 | 3.0 | 3.50 | ACR |
|---|---|---|---|---|---|---|---|---|---|---|---|---|---|
| COHEN | FS | FS | 66.6 | 58.2 | 49.0 | 42.4 | 37.4 | 27.8 | 19.4 | 14.4 | 12.0 | 8.6 | 1.098 |
| COHEN-DS | FS | DS | **67.8** | 61.4 | 53.6 | 45.6 | **42.0** | 30.4 | 23.4 | 18.8 | 14.6 | 10.2 | 1.257 |
| COHEN-DS$^2$ | DS | DS | 67.4 | **64.2** | **58.4** | **47.4** | 41.8 | **31.8** | **25.0** | **21.2** | **17.2** | **11.0** | **1.319** |

Figure 4: **Certified accuracy comparison against SMOOTHADV.** We compare SMOOTHADV against SMOOTHADV-DS and SMOOTHADV-DS$^2$. We show CIFAR10 and ImageNet results in the first and second rows, respectively. The last column shows the envelope across $\sigma$.

3 shows that COHEN-DS is superior to COHEN in certified accuracy across almost all radii and for all training $\sigma$ on both datasets. This is also evident from the envelope plots in the last column of Figure 3. In Table 1, we report the best certified accuracy per radius over all training $\sigma$ for COHEN (envelope figure) against our best COHEN-DS, cross-validated over all training $\sigma$ and the number of iterations in Algorithm 1 $K$, accompanied with the corresponding ACR score. For instance, we observe that data dependent certification COHEN-DS can significantly boost certified accuracy at radii 0.5 and 0.75 by 7.7% (from 40.1 to 47.8) and 9.1% (from 29.2% to 38.3%), respectively, and by 0.193 ACR points on CIFAR10. Moreover, we boost the certified accuracy on ImageNet by 4.6% and 3.2% at 0.5 and 0.75 radii, respectively, and by 0.159 ACR points.

**DS for training and certification.** We employ data dependent smoothing in both training and certification for COHEN models (denoted as COHEN-DS$^2$) by running Algorithm 2. For CIFAR10, we train COHEN first with fixed $\sigma$ for 50 epochs, *i.e.* $K = 0$ in Algorithm 1, and then we perform data dependent smoothing with $K = 1$ for the remaining 150 epochs. For ImageNet experiments, we only finetune the provided models for 30 epochs using Algorithm 2 with $K = 1$. Once training is complete, we certify all trained models with Algorithm 1 using the memory based certification. In Figure 3, we observe that COHEN-DS$^2$ can further improve certified accuracy across all trained models on both CIFAR10 and ImageNet. This is also evident in the last column of Figure 3 that shows the best certified accuracy per radius (envelope) over all training $\sigma$. We note that COHEN-DS$^2$ improves the certification accuracy of COHEN-DS by 2.6% and by 0.9% at radii 0.5 and 0.75 respectively on CIFAR10, and by 4.8% and 1.8% at radii 0.5 and 0.75 respectively on ImageNet. The improvements are consistently present over a wide range of radii on both datasets. We do observe that the ACR score for COHEN-DS$^2$ on CIFAR10 marginally drops compared to COHEN-DS. We believe that this is due to the fact that some inputs that are classified correctly at the small radii have an overall larger certification radius for COHEN-DS compared to COHEN-DS$^2$ on CIFAR10. Regardless, COHEN-DS$^2$ substantially outperforms COHEN by 0.173 ACR points. As compared to COHEN-DS, COHEN-DS$^2$ improves the ACR on ImageNet from 1.257 to 1.319.

## 4.2 SMOOTHADV + DS

We combine our data dependent smoothing strategy with the more effective SMOOTHADV, which trains the smoothed classifier for every $x$ on the adversarial example $\hat{x}$ that maximizes $-\log \mathbb{E}_{\epsilon \sim \mathcal{N}(0, \sigma^2 I)} \left[ f_\theta^y(x' + \epsilon) \right]$, where $\|x' - x\| \leq \zeta$. For CIFAR10 experiments, we follow the train-

Table 2: **Best certified accuracy per radius and ACR** of SMOOTHADV, SMOOTHADV-DS and SMOOTHADV-DS$^2$.

| CIFAR10 | Radius Train | Certify | 0.0 | 0.25 | 0.50 | 0.75 | 1.00 | 1.25 | 1.50 | 1.75 | 2.00 | 2.25 | ACR |
|---|---|---|---|---|---|---|---|---|---|---|---|---|---|
| SMOOTHADV | FS | FS | 76.0 | 62.4 | 46.7 | 34.6 | 26.5 | 19.5 | 12.9 | 7.5 | 0.0 | 0.0 | 0.681 |
| SMOOTHADV-DS | FS | DS | 75.7 | 66.4 | 52.1 | 38.8 | 30.6 | 22.2 | 15.0 | 8.5 | 4.2 | 1.8 | 0.799 |
| SMOOTHADV-DS$^2$ | DS | DS | **76.2** | **66.8** | **52.8** | **39.3** | **30.8** | **22.6** | **15.1** | **8.8** | **4.3** | **2.0** | **0.812** |

| ImageNet | Radius Train | Certify | 0.0 | 0.25 | 0.50 | 0.75 | 1.00 | 1.50 | 2.0 | 2.5 | 3.0 | 3.50 | ACR |
|---|---|---|---|---|---|---|---|---|---|---|---|---|---|
| SMOOTHADV | FS | FS | 60.8 | 57.8 | 54.6 | 50.4 | 42.2 | 35.6 | 25.6 | 20.4 | 18.0 | 14.2 | 1.287 |
| SMOOTHADV-DS | FS | DS | 62.0 | 60.4 | 57.4 | 53.2 | 47.0 | 39.2 | 29.2 | 23.8 | 19.6 | 15.2 | 1.445 |
| SMOOTHADV-DS$^2$ | DS | DS | **62.2** | **60.6** | **58.8** | **54.2** | **48.2** | **43.0** | **30.6** | **25.4** | **21.6** | **18.6** | **1.514** |

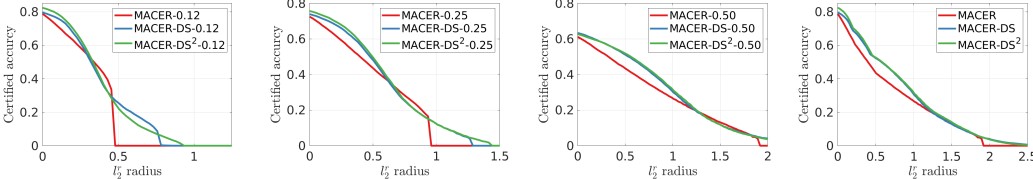

Figure 5: **Certified accuracy comparison against MACER.** We compare MACER against MACER-DS and MACER-DS$^2$ for several $\sigma$ on CIFAR10. The last column shows the envelope.

ing procedure of SMOOTHADV, where the adversary $\hat{x}$ is computed with 2 PGD (proximal gradient descent) steps with $\zeta = 0.25$ and one augmented sample to estimate the expectation. For ImageNet experiments, we use the best reported models, in terms of certified accuracy, provided by the authors, which correspond to $\zeta = 0.5$ for $\sigma = 0.25$ and $\zeta = 1.0$ for $\sigma \in \{0.5, 1.0\}$.

**DS for certification only.** Similar to COHEN, we first certify SMOOTHADV models trained with the same fixed $\sigma$. Then, we certify the proposed data dependent $\sigma_x^*$ models using the memory-based certification, which we refer to as SMOOTHADV-DS. In Figure 4, we show the certified accuracy for both CIFAR10 and ImageNet in the first and second rows, respectively. The last column shows the envelopes per radius. Even though they both share the same classifier $f_\theta$, SMOOTHADV-DS significantly improves upon SMOOTHADV over all radii and all values of $\sigma$ in training for both CIFAR10 and ImageNet. In particular, for models trained with $\sigma = 0.25$, SMOOTHADV achieves a zero certified accuracy for large certification radii ($\geq 1.0$), while SMOOTHADV-DS achieves non-trivial certified accuracy in these cases. Similar to the earlier setup, we report the best certified accuracy along with the ACR scores in Table 2. We improve over SMOOTHADV by large margins. For example, the certified accuracy at $0.5$ radius increases by $5.4\%$ and $2.8\%$ on CIFAR10 and Imagenet, respectively. The improvement is consistent over all radii. The ACR also improves by $0.118$ and $0.158$ on CIFAR10 and ImageNet, respectively.

**DS for training and certification.** We fine tune the SMOOTHADV trained models (either the retrained CIFAR10 models or the ImageNet models provided by SMOOTHADV) using Algorithm 2, where $\sigma_x^*$ is computed using Algorithm 1. We report the per $\sigma$ certification accuracy comparing SMOOTHADV-DS$^2$ (certified also using memory based certification) to both SMOOTHADV-DS and SMOOTHADV. SMOOTHADV-DS$^2$ further improves the certified accuracy as compared to SMOOTHADV-DS with performance gains more prominent on ImageNet. While the improvement of SMOOTHADV-DS$^2$ over SMOOTHADV-DS is indeed small, *e.g.* $0.7\%$ at radius 0.5 on CIFAR10, we observe that the performance gaps are much larger on ImageNet reaching $1.4\%$ at 0.5 radius as shown in Table 2. We see a similar trend in ACR with improvements of $0.013$ and $0.069$ on CIFAR10 and ImageNet, respectively. SMOOTHADV-DS$^2$ boosts the certified accuracy of SMOOTHADV at radius 0.5 by $6.1\%$ and $4.2\%$ on CIFAR10 and ImageNet, respectively.

### 4.3 MACER + DS

We integrate data dependent smoothing within MACER which trains $g_\theta$ by minimizing over the parameters $\theta$ the following objective $-\log g_\theta(x) + \frac{\lambda\sigma}{2} \max\left(\gamma - \frac{2R}{\sigma}, 0\right) . \mathbb{1}\{\arg\max_c g_\theta^c(x) = y\}$. where $R$ also depends on $\theta$. While this seems to be similar in spirit to our approach, we in fact maximize the certification radius over $\sigma$ with fixed parameters $\theta$ for every $x$. We conduct experiments

Table 3: **Best certified accuracy per radius and ACR** of MACER, MACER-DS and MACER-DS$^2$.

| CIFAR10 | Radius Train | Certify | 0.0 | 0.25 | 0.50 | 0.75 | 1.00 | 1.25 | 1.50 | 1.75 | 2.00 | 2.25 | ACR |
|---|---|---|---|---|---|---|---|---|---|---|---|---|---|
| MACER | FS | FS | 78.8 | 59.3 | 43.6 | 34.7 | 26.6 | 19.4 | 13.0 | 7.50 | 0.0 | 0.0 | 0.702 |
| MACER-DS | FS | DS | 79.5 | 66.7 | 52.3 | 43.0 | 30.8 | 19.5 | 12.8 | 7.55 | **3.97** | **1.67** | **0.841** |
| MACER-DS$^2$ | DS | DS | **82.4** | **68.3** | **52.7** | **43.5** | **31.7** | **20.6** | **13.8** | **7.92** | 3.65 | 1.39 | 0.807 |

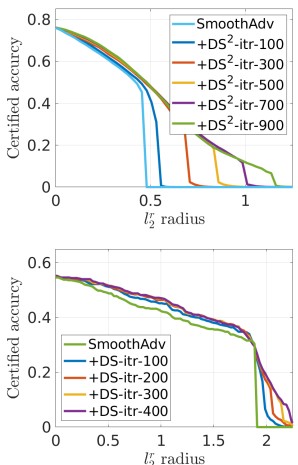

Figure 6: **Qualitative examples of estimated $\sigma_x^*$ for different inputs.** From left to right: the set of three images on the left: clean image, fixed $\sigma = 0.5$, and estimated $\sigma_x^* = 0.368$ that maximizes the certification radius. Similarly for second set but with $\sigma = 0.25$ and $\sigma_x^* = 0.423$. This demonstrates that $\sigma_x^*$, which maximizes the certification radius, can vary significantly w.r.t. input $x$.

on CIFAR10[3] following the training procedure of MACER estimating the expectation with 64 samples, $\lambda = 12$, and $\gamma = 8$. We set $n = 8$ in Algorithm 1 with ablations on $n = 1$ in the **appendix**.

**DS for certification only.** Similar to the earlier setup in COHEN and SMOOTHADV, we certify models with fixed $\sigma$ and then with data dependent $\sigma_x^*$ using the memory based certification, referred to as MACER-DS. In Figure 5, we observe that MACER-DS significantly outperforms MACER particularly in the large radius region. This can also be seen in the envelope figure reporting the best certified accuracy per radius over $\sigma$. Similarly, Table 3 demonstrates the benefits of data dependent smoothing, where it boosts certified accuracy by 7.4% (from 59.3% to 66.7%) and 8.7% (43.6 to 52.3) at 0.25 and 0.5 radii, respectively. Moreover, we improve ACR by 0.139 points.

**DS for training and certification.** We incorporate data dependent smoothing as part of MACER training and certification in a similar fashion to the earlier setup, dubbed MACER-DS$^2$. Figure 5 shows the improvement of MACER-DS$^2$ over the certification only MACER-DS over all trained models. Table 3 summarizes the best certified accuracy per radius. Overall, we find that the performance is comparable or slightly better than MACER-DS, which is still significantly better than MACER by 8.67% at radius 0.5. We also observe that MACER-DS enjoys better ACR than MACER-DS$^2$ with both being far better than the MACER baseline.

## 4.4 DISCUSSION AND ABLATION

**Varying $K$.** We pose the question: does attaining better solutions to our proposed Objective 3 improve certified accuracy? To answer this, we control the solution quality of $\sigma_x^*$ by certifying trained models with a varying number of stochastic gradient ascent iterations $K$ in Algorithm 1. In particular, we certify the trained models SMOOTHADV-DS$^2$ and SMOOTHADV-DS on CIFAR10 and ImageNet, respectively, with a varying $K$. We leave the rest of the experiments for other models to the **appendix**. We observe in Figure 7 that the certified accuracy per radius consistently improves as $K$ increases, particularly in the large radius regime. This is expected, since Algorithm 1 produces better optimal smoothing $\sigma_x^*$ per input $x$ with larger $K$, which in turn improves the certification radius leaving room for improvements with more powerful optimizers[4].

**Visualizing $\sigma_x^*$.** We show the variation of $\sigma_x^*$ that maximizes the certification radius over different inputs $x$. Figure 6 shows two examples, where the first and fourth columns contain the clean images. In the second column, a choice of fixed $\sigma = 0.5$ is too large compared to our estimated $\sigma_x^* = 0.368$ that maximizes the certification radius as per Algorithm 1. As for the fifth column, we observe that a constant $\sigma = 0.25$ is far less than $\sigma_x^* = 0.423$. This indicates that indeed the $\sigma_x^*$ maximizing the certification radius varies significantly over inputs.

Figure 7: **Varying $K$ in Algorithm 2.** Top figure shows certification with $\sigma_0 = 0.12$ on CIFAR10 and $\sigma_0 = 0.5$ on ImageNet is shown at the bottom.

---

[3]ResNet-50 trained models on ImageNet are not provided by the authors.

[4]Greedy heuristics solving Equation 3 are in the **appendix**, as they perform far worse than our approach.

## 5 REPRODUCIBILITY STATEMENT

This work has two main components to reproducing our experimental results; Algorithm 1 and the memory-based certification. In the setup paragraph in our experimental section, we mentioned all necessary implementation details for Algorithm 1. Nonetheless, we provide an implementation using PyTorch (Paszke et al., 2019) of Algorithm 1 in Appendix A. As for memory based certification, we provide a detailed algorithm supported with PyTorch implementation of our memory-based certification in Appendix C. Algorithm 2 is based on publically available code training with links in the appendix. Moreover, since we used trained models with different frameworks, we provide the links to the corresponding Github public repositories in Appendix A. We also attach code reproducing some of the experimetns in the zip file attached in the submission. The full code will be released in the future.

## 6 ETHICS STATEMENT

Since our work has an experimental nature where all of our experiments are computational, we believe that our work did not raise any ethical concerns. In particular, we did not include any human subjects, any potential harmful insights, nor discrimination.

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

## A    IMPLEMENTATION DETAILS

For reproducibility, we will release the full code upon acceptance. Nevertheless, we give the detailed implementation of Algorithm 1 in PyTorch Paszke et al. (2019) below.

```
1  import torch
2  from torch.autograd import Variable
3  from torch.distributions.normal import Normal
4  def OptimzeSigma(model, batch, alpha, sig_0, K, n):
5      device='cuda:0'
6      batch_size = batch.shape[0]
7
8      sig = Variable(sig_0, requires_grad=True).view(batch_size, 1, 1, 1)
9      m = Normal(torch.zeros(batch_size).to(device), torch.ones(batch_size)
       .to(device))
10
11     #Reshaping for n > 1
12     new_shape = [batch_size * n]
13     new_shape.extend(batch_size)
14     new_batch = batch.repeat((1,n, 1, 1)).view(new_shape)
15     sigma_repeated = sig.repeat((1, n, 1, 1)).view(-1,1,1,1)
16
17     for _ in range(K):
18         eps = torch.randn_like(new_batch)*sigma_repeated #
       Reparamitrization trick
19         out = model(new_batch + eps).reshape(batch_size, n, 10).mean(1) #
       10 for CIFAR10
20
21         vals, _ = torch.topk(out, 2)
22         vals.transpose_(0, 1)
23         gap = m.icdf(vals[0].clamp_(0.02, 0.98)) - m.icdf(vals[1].clamp_
       (0.02, 0.98))
24         radius = sig.reshape(-1)/2 * gap  # The radius formula
25         grad = torch.autograd.grad(radius.sum(), sig)
26
27         sig.data += alpha*grad[0]  # Gradient Ascent step
28
29     return sig.reshape(-1)
```

For comparisons against Cohen et al. (2019), we followed their official code in https://github.com/locuslab/smoothing. We also followed the common practice in using their provided code for certifying all models in all of our experiments. For comparisons against *SmoothAdv* (Salman et al., 2019a), we also followed their official implementation in https://github.com/Hadisalman/smoothing-adversarial and similarly for *MACER* https://github.com/RuntianZ/macer.

## B    DATA DEPENDENT GREEDY SEARCH OVER $\sigma$

We observe that the optimization problem equation 3 that we solve for every input $x$ is one dimensional in $\sigma_x^*$. In this section, we show that heuristic grid search procedures are far inferior to solving Equation equation 3 with our solver in Algorithm 1. In particular, we show that under the same sample complexity as our approach for data dependent certification a trivial heuristic grid search as a baseline does not work. We conduct experiments where we only certify with data dependent smoothing a pre-trained model SmoothAdv with training $\sigma \in \{0.12, 0.25.0.50\}$ on CIFAR10. We examine a single model SmoothAdv-DS which is certified with $K = 100$ iterations and with $n = 1$ to approximate the expectation in Algorithm 1. Observe that since $n = 1$, and including the forward and backward passes computation, our data dependent certification of SmoothAdv-DS has a total of 200, since $K = 100$, evaluations for every given $x$ before performing the certification with the optimized $\sigma_x^*$. To that end, we compare against a crude grid search baseline over $\hat{\sigma}_x^*$, and for a fair comparison, with a total of 200 evaluations. We restrict the grid search to $\hat{\sigma}_x^* \in [0, 1]$ with a resolution of $n/200$ so that the total number of evaluations is always exactly 200 similar to our SmoothAdv-DS. That is to say, the grid heuristic search solves the following problem:

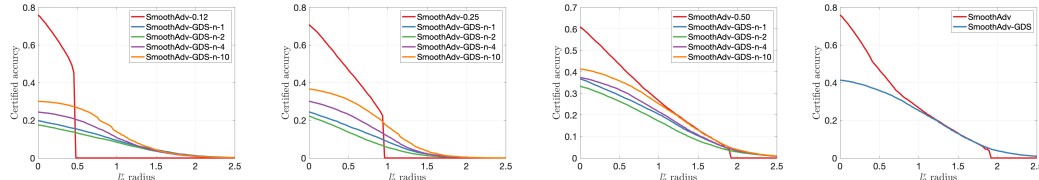

Figure 8: **Certified accuracy per radius per $\sigma$ comparison of SmoothAdv against greedy search heuristic data dependent smoothing on CIFAR10.** We observe that when the number of samples $n$ in Equation 4 the better the performance. Moreover, and despite that the total number of evaluations of the data dependent certification matches our approach SmoothAdv-DS reported in Figure 4, the performance is still far inferior to even data dependent certification as also evident from the last envelope figure.

$$
\hat{\sigma}_x^* = \underset{\sigma_i \in \{0, \frac{n}{200}, \frac{2n}{200}, \dots, 1-\frac{n}{200}, 1\}}{\arg\max} \frac{\sigma_i}{2} \left( \Phi^{-1}\left( \frac{1}{n} \sum_{i=1}^{n} \hat{f}_\theta^{c_A}(x+\sigma_i\hat{\epsilon})] \right) - \Phi^{-1}\left( \max_{c \neq c_A} \frac{1}{n} \sum_{i=1}^{n} \hat{f}_\theta^{c}(x+\sigma_i\hat{\epsilon})] \right) \right).
\tag{4}
$$

We also explore with the number of samples to $n \in \{1, 2, 4, 10\}$ for the grid search pipeline. Note that this trades-off the accuracy of the expectation approximation to the resolution of the solution $\hat{\sigma}_x^*$.

We summarize our results in Figure 8. Note that in the first three figures, we report certified accuracies for when the model is certified with the same $\sigma = \{0, 12, 0.25, 0.50\}$ used in training without data dependent smoothing, i.e. fixed $\sigma$ for all inputs. We refer to these plots as SmoothAdv-0.12, SmoothAdv-0.25 and SmoothAdv-0.50. In addition, we refer to the data dependent baseline grid search heuristic as SmoothAdv-GDS-n-1, SmoothAdv-GDS-n-2, SmoothAdv-GDS-n-4, and SmoothAdv-GDS-n-10 where $n$ refers to the number of samples approximating the expectation in Equation equation 4. We report the envelops in the last figure.

At first we observe that the larger $n$ used to approximate the expectation, the better the overall certification accuracy. This is regardless of the $\sigma$ used to train the model. However, the performance is still far inferior to the baseline that is data *independent* which is inferior to our approach. This is also evident from the envelope last figure. This indicates that while data dependent smoothing is essential towards improving performance, a careful optimization is necessary for it to work. We reiterate here that both the grid search heuristic and our approach use the same number of evaluations, *i.e.* 200, when certifying the model; however, our approach reported in Figure 4 are far more superior.

## C   MEMORY-BASED CERTIFICATION FOR DATA DEPENDENT CLASSIFIERS

---
**Algorithm 3:** Memory-Based Certification

**Input:** input point $x_{N+1}$, certified region $\mathcal{R}_{N+1}$, prediction $\mathcal{C}_{N+1}$, and memory $\mathcal{M}$
**Result:** Prediction for $x_{N+1}$ and certified region at $x_{N+1}$ that does not intersect with any
      certified region in $\mathcal{M}$.
**for** $(x_i, \mathcal{C}_i, \mathcal{R}_i) \in \mathcal{M}$ **do**
    **if** $\mathcal{C}_{N+1} \neq \mathcal{C}_i$ **then**
        **if** $x_{N+1} \in \mathcal{R}_i$ **then**
            $\tilde{\mathcal{R}}_{N+1} = \texttt{LargestInSubset}(\mathcal{R}_i, \mathcal{R}_{N+1})$,
            $\mathcal{R}_{N+1} \leftarrow \tilde{\mathcal{R}}_{N+1}$;
            $\mathcal{C}_{N+1} \leftarrow \mathcal{C}_i$
        **else if** $\texttt{Intersect}(\mathcal{R}_{N+1}, \mathcal{R}_i)$ **then**
            $\mathcal{R}'_{N+1} = \texttt{LargestOutSubset}(\mathcal{R}_i, \mathcal{R}_{N+1})$;
            $\mathcal{R}_{N+1} \leftarrow \mathcal{R}'_{N+1}$;
**end**
add $(x_{N+1}, \mathcal{C}_{N+1}, \mathcal{R}_{N+1})$ to $\mathcal{M}$;
**return** $\mathcal{C}_{N+1}, \mathcal{R}_{N+1}$;

---

Let $\mathcal{M} = \{(x_i, \mathcal{C}_i, \mathcal{R}_i)\}_{i=1}^N$ be set of the triplets: the input $x_i$, the prediction of $x_i$ denoted by $\mathcal{C}_i$ and the certification region at $x_i$ denoted as $\mathcal{R}_i$ which is characterized by the certification radius $R_i$ and the center $x_i$. Moreover, we assume that $\mathcal{R}_i \cap \mathcal{R}_j = \emptyset, \forall i \neq j, \mathcal{C}_i \neq \mathcal{C}_j$. That is to say, none of the certification regions of the inputs stored in the memory $\mathcal{M}$ intersect for inputs with different predictions. This is the key property for a sound certification procedure. Otherwise, if such a property does not hold, then this implies that the data dependent classifier produces different predictions within the same certified region. In what follows, and to circumvent this nuisance in the data dependent classifier $g_\theta$, we rely on updating the memory while enforcing this property to hold. In particular, we certify the data dependent classifier using the classical Monte Carlo approach by Cohen et al. (2019) while guaranteeing that the certified regions does not intersect with the certification region of any previously predicted inputs.

In what follows, we present Algorithm 3 that enforces the non-intersection property of certified regions $\mathcal{M}$. Let $\mathcal{R}_{N+1}$ be the certified region at $x_{N+1}$ of $g_\theta$. **(i)** If $x_{N+1} \in \mathcal{R}_i$ and $\arg\max_c g_\theta^c(x_{N+1}) \neq \mathcal{C}_i$, we find the largest $\tilde{\mathcal{R}}_{N+1}$ such that the following two properties hold $\tilde{\mathcal{R}}_{N+1} \subset \mathcal{R}_{N+1}$ and $\tilde{\mathcal{R}}_{N+1} \subset \mathcal{R}_i$. For when the certified regions $\mathcal{R}_i$ are simple $\ell_2$-balls, finding the largest $\tilde{\mathcal{R}}_{N+1}$ satisfying previous two properties is straightforward. We denote this with the function `LargestInSubset` (second example in Figure 2). We then update $\mathcal{R}_{N+1}$ with the refined $\tilde{\mathcal{R}}_{N+1}$ and change $\mathcal{C}_{N+1}$ to $\mathcal{C}_i$. **(ii)** Otherwise, if $\mathcal{R}_{N+1} \cap \mathcal{R}_i \neq \emptyset$ where $x_{N+1} \notin \mathcal{R}_i$ and $\arg\max_c g_\theta^c(x_{N+1}) \neq \mathcal{C}_i$, we find $\mathcal{R}'_{N+1}$ such that $\mathcal{R}'_{N+1} \subseteq \mathcal{R}_{N+1}$ is the largest subset of $\mathcal{R}_{N+1}$ non-intersecting with $\mathcal{R}_i$. We denote this function `LargestOutSubset` (third example in Figure 2). We then update $\mathcal{R}_{N+1}$ with the refined $\mathcal{R}'_{N+1}$. Moreover, computing `LargestOutSubset` for when $\mathcal{R}_i$ are $\ell_2$-balls is straightforward. At last, note that `Intersect` is a function that returns whether two $\ell_2$-balls intersect. At last, we then add $(x_{N+1}, \mathcal{C}_{N+1}, \mathcal{R}_{N+1})$ to memory. We provide below a pytorch implementation of the memory-based certification of the pseudo-algorithm 3.

While the memory-based certification is essential for a sound certification, empirically on CIFAR10 and ImageNet, we never found in any of the experiments a case where two inputs predicted differently suffer from intersecting certified regions. That is to say, the certified regions in the memory for every input is the certified regions granted by the Monte Carlo certificates of Cohen et al. (2019) for the data dependent classifier. We hypothesize that this is due to the following reasons: **(i)** Image datasets have very high dimensionality, resulting in samples very far apart, compared with the certified radius that randomized smoothing could provide. Thus, it is very unlikely to find two samples that have intersecting certified regions. **(ii)** Even if the rare case where two image inputs are close to one another that their certified regions intersect, we found that the data dependent classifier $g_\theta$ predicts these inputs similarly (the left example of Figure 2). This is since the data dependent classifier is trained to output smooth prediction, *i.e.* prediction changes are small for small input changes, resulting in a shared prediction. **(iii)** To maintain reasonable test accuracy on clean samples, the values of $\sigma$, and correspondingly optimized $\sigma_x^*$ used in smoothing are moderately low ($\sigma_x^* \leq 1.0$). This results in limited smaller certified regions, $\ell_2$ balls of radii $\approx 4\sigma_x^*$ which is much smaller than the distance between inputs in higher dimensional data (*e.g.* ImageNet).

It is worthwhile mentioning that while the memory-based certificate could work, in principle, independently without being combined with the data dependent smooth classifier under any arbitrary choice of a certification radius for every input; this results in a sub-optimal certification. This is since this may result in one of the following situations: **(i)** Assigning large radii for every input will yield a classifier that is very robust, but inaccurate. This is since several new points to be certified later will more likely fall in the certification region requiring either changing their prediction (inaccurate predictions) or reducing their certification radius. Therefore, measuring the certified accuracy for such a classifier will be very poor since it counts for both accuracy and robustness. **(ii)** Assigning, on the other hand, small certified radii for every input will result in a highly accurate classifier but very low robustness. Hence, this also results in a very small certified accuracy at large radii. Therefore, we combine the memory based certificate with our data-dependent smooth classifier that has a better robustness/accuracy tradeoff.

For completeness, we provide the full implementation of the memory-based algorithm using PyTorch.

```python
import torch
class Memory_Based_Certification(object):
    def __init__(self):
        self.saved_radii = []
```

```
 5          self.saved_images = []
 6          self.saved_predictions = []
 7
 8
 9      def _internal_adjustment(self, img, rad, pre):
10          diff = torch.norm(img.reshape(1, -1) - torch.stack(self.
    saved_images).reshape(len(self.saved_radii), -1), dim=1)
11
12          where_overlap =  diff < (torch.tensor(self.saved_radii) + rad)
13          #Check whether this image is with overlap with any other
    instances
14          if where_overlap.any():
15              preds_overlap = self.saved_predictions[where_overlap]
16              where_overlap_diff_class = preds_overlap != pre
17
18              #Check whether this image is with overlap with instances with
     different prediction
19              if where_overlap_diff_class.any():
20                  #Get the radii, differences where the overlap
21                  saved_radii_with_overlap = self.saved_radii[where_overlap
    ]
22                  dif_with_overlap = diff[where_overlap]
23
24                  preds_overlap_with_diff_class = preds_overlap[
    where_overlap_diff_class]
25                  rad_with_overlap_diff_class = saved_radii_with_overlap[
    where_overlap_diff_class]
26                  dif_with_overlap_diff_class = dif_with_overlap[
    where_overlap_diff_class]
27
28                  rad, rad_idx = torch.min(dif_with_overlap_diff_class -
    rad_with_overlap_diff_class)
29
30                  if rad.item() < 0:
31                      pre = preds_overlap_with_diff_class[rad_idx]
32
33                  rad = torch.abs(rad).item()
34          return rad, pre
35
36      def adjust_radius(self, img, rad, pre):
37          #The img already exists in the saved dictionary
38          if img in self.saved_images:
39              idx = self.saved_images == img
40              return self.saved_radii[idx], self.saved_predictions[idx]
41
42          if self.saved_radii != []: #Saved dictionaries are not empty
43              rad, pre = self._internal_adjustment(img, rad, pre)
44
45          self.saved_radii.append(rad)
46          self.saved_images.append(img)
47          self.saved_predictions.append(pre)
48          return rad, pre
```

## D  ADDITIONAL VISUALIZATIONS

Here, we show similar results to the one in Figure 6. Similar to the earlier observations, while model parameters are fixed, optimal smoothing parameters vary per sample.

## E  LIMITATIONS, BROADER IMPACT AND COMPUTE POWERS USED.

**Limitations.**  Similar to any certification framework, the main limitation of this kind of work is its running time to compute the certified radius. The proposed memory-based certification is at

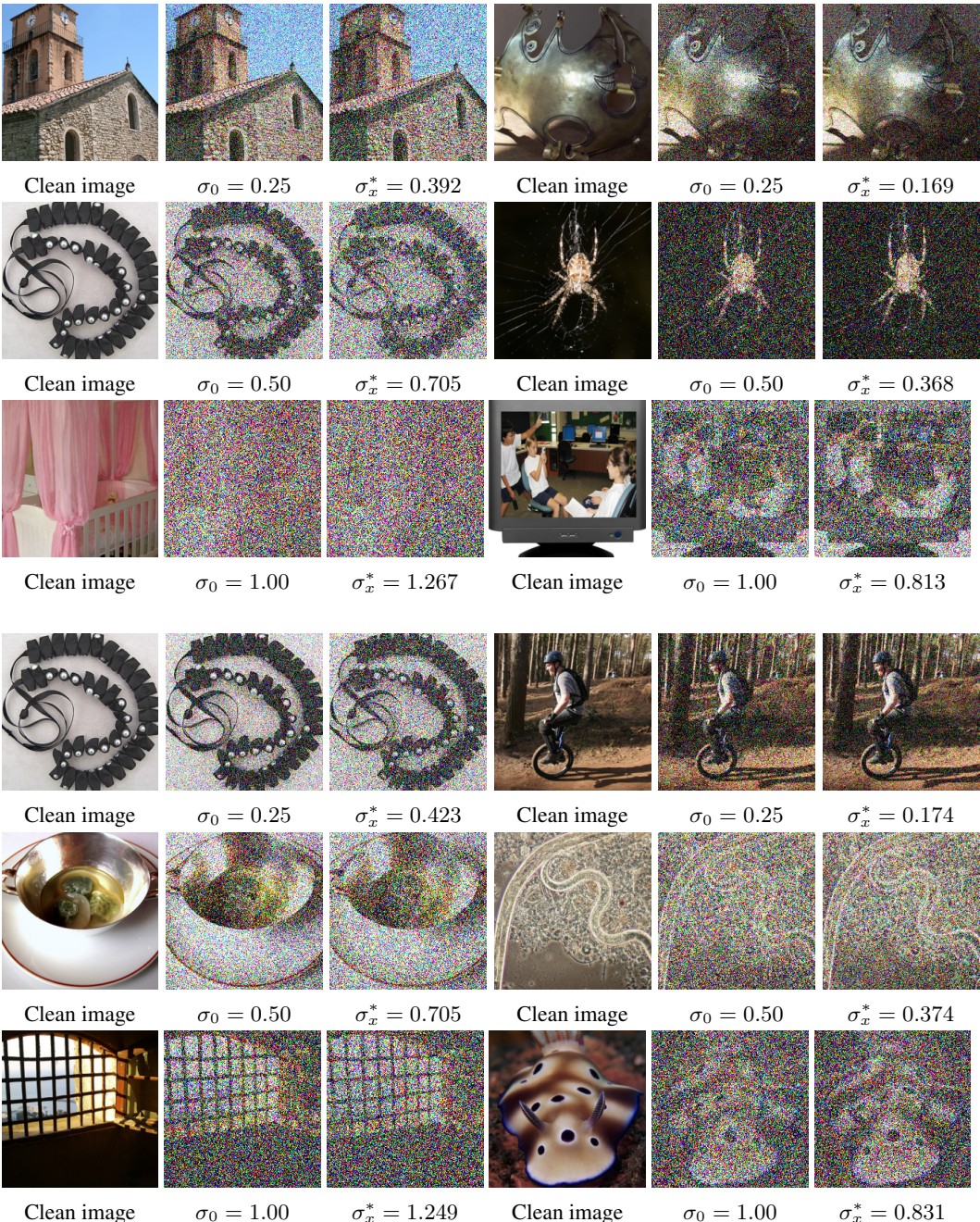

Figure 10: **Visualizing the extreme $\sigma$.** We report the visual comparison between the constant $\sigma_0$ and the optimal attained $\sigma_x^*$ for Cohen Models (first three rows) and SmoothAdv (last three rows) both on ImageNet.

the cost of both memory and computational complexity. While the memory cost is of order $\mathcal{O}(N)$ the computational complexity is more involved. Let $p$ be the probability that a new point $x_{N+1}$ be in one of the certification regions $\mathcal{R}_i$, $n$ is the complexity of computing a certification region at $x_{N+1}$, *i.e.* $\mathcal{R}_{N+1}$, using the classical Monte Carlo Algorithms of Cohen et al. (2019), then the expected computational complexity of prediction or certification will be $\mathcal{O}(Np + (1-p)(2N+n))$. The factor $2N + n$ is due to performing $N$ comparisons to check that $x_{N+1}$ is not in any $\mathcal{R}_i$, then computing $\mathcal{R}_{N+1}$ of complexity $n$, and at last a complexity of $N$ for computing $\mathcal{R}'_{N+1}$. Informally, for larger $N$ and in small dimensional input, $p \approx 1$ leading to a complexity of order $N$. When $N$

is smaller compared to the input dimension, we have $p \approx 0$ with an expected complexity of order $2N + n$. Moreover, the memory-based data dependent smooth classifier is order dependent. That is, the certified accuracy depends on the order at which the data at test time is presented. However, this is not the case if there is no overlap between the certified regions of differently predicted inputs (middle and right scenarios of Figure 2 do not occur). We found that is the case in all of our experiments making our memory-enhanced data dependent smooth classifier order invariant. We elaborated on why we believe that is the case in Appendix C. We plan in future extension to delve into more practical solution to this problem. We postpone the design for more efficient algorithms that validate the soundness of data dependent certification to future work.

**Broader Impact.** While the performance of Deep Neural networks is dominating over several fields, the existence of adversarial examples hinders their deployment in lots of applications. This raise the attention to build networks that are not only accurate, but also robust to such perturbations. This work takes a step towards a remedy for this nuisance by improving the certified robustness of deep neural networks.

**Compute Powers.** In our experiment on CIFAR10, we used either NVIDIA Quadro RTX-600 GPU or NVIDIA 1080TI GPU. For ImageNet experiments, we used NVIDIA-V100 GPU. Note that one GPU was enough to run any of our experiments.

## F    DETAILED ABLATIONS

### F.1    COHEN VS COHEN-DS VS COHEN-DS$^2$

In this section, we detail the certified accuracy per radius for all trained models per $\sigma$ for Cohen and per $\sigma$ and number of iterations $K$ for Cohen et al. (2019), Cohen-DS and Cohen-DS$^2$ in Algorithm 1 on both CIFAR10 and ImageNet.

### F.2    SMOOTHADV VS SMOOTHADV-DS VS SMOOTHADV-DS$^2$

In a similar spirit to the previous section, we report the certified accuracy for the SmoothAdv variants, namely, SmoothAdv Salman et al. (2019a), SmoothAdv-DS and SmoothAdv-DS$^2$ on CIFAR10 and ImageNet.

Table 4: **Certified accuracy per radius on CIFAR10.** We compare Cohen against Cohen-DS under varying $\sigma$ and number of iterations $K$ in Algorithm 1.

| | $\ell_2^r$ (CIFAR10) | | 0.0 | 0.25 | 0.50 | 0.75 | 1.00 | 1.25 | 1.50 | 1.75 | 2.00 | 2.25 | 2.50 |
|---|---|---|---|---|---|---|---|---|---|---|---|---|---|
| Cohen | $\sigma = 0.12$ | | 79.89 | 56.26 | 0.0 | 0.0 | 0.0 | 0.0 | 0.0 | 0.0 | 0.0 | 0.0 | 0.0 |
| | $\sigma = 0.25$ | | 74.45 | 58.34 | 40.13 | 22.85 | 0.0 | 0.0 | 0.0 | 0.0 | 0.0 | 0.0 | 0.0 |
| | $\sigma = 0.50$ | | 63.72 | 52.15 | 40.13 | 29.17 | 20.18 | 13.08 | 7.33 | 3.33 | 0.0 | 0.0 | 0.0 |
| Cohen-DS | $\sigma = 0.12$ | K=100 | 77.19 | 61.27 | 20.8 | 5.47 | 1.23 | 0.02 | 0.0 | 0.0 | 0.0 | 0.0 | 0.0 |
| | $\sigma = 0.12$ | K=200 | 74.98 | 60.67 | 19.75 | 4.05 | 0.94 | 0.22 | 0.01 | 0.0 | 0.0 | 0.0 | 0.0 |
| | $\sigma = 0.12$ | K=300 | 73.56 | 60.08 | 19.63 | 4.37 | 1.12 | 0.36 | 0.08 | 0.0 | 0.0 | 0.0 | 0.0 |
| | $\sigma = 0.12$ | K=400 | 72.11 | 59.38 | 19.58 | 4.27 | 1.39 | 0.58 | 0.13 | 0.0 | 0.0 | 0.0 | 0.0 |
| | $\sigma = 0.12$ | K=500 | 70.78 | 58.77 | 19.5 | 4.77 | 1.51 | 0.68 | 0.14 | 0.0 | 0.0 | 0.0 | 0.0 |
| | $\sigma = 0.12$ | K=600 | 70.17 | 58.46 | 19.51 | 4.75 | 1.63 | 0.85 | 0.18 | 0.03 | 0.0 | 0.0 | 0.0 |
| | $\sigma = 0.12$ | K=700 | 69.83 | 58.25 | 19.91 | 5.05 | 1.83 | 0.88 | 0.21 | 0.04 | 0.0 | 0.0 | 0.0 |
| | $\sigma = 0.12$ | K=800 | 69.25 | 57.97 | 19.75 | 5.04 | 1.99 | 0.95 | 0.17 | 0.03 | 0.0 | 0.0 | 0.0 |
| | $\sigma = 0.12$ | K=900 | 68.27 | 57.51 | 19.91 | 5.07 | 1.94 | 0.93 | 0.21 | 0.04 | 0.0 | 0.0 | 0.0 |
| | $\sigma = 0.25$ | K=100 | 73.17 | 64.54 | 47.48 | 22.58 | 6.53 | 1.82 | 0.47 | 0.0 | 0.0 | 0.0 | 0.0 |
| | $\sigma = 0.25$ | K=200 | 71.62 | 64.2 | 47.3 | 21.66 | 5.45 | 1.15 | 0.34 | 0.13 | 0.06 | 0.01 | 0.0 |
| | $\sigma = 0.25$ | K=300 | 70.23 | 63.91 | 47.44 | 21.75 | 5.38 | 1.37 | 0.43 | 0.21 | 0.13 | 0.04 | 0.02 |
| | $\sigma = 0.25$ | K=400 | 69.41 | 63.42 | 47.43 | 22.23 | 5.83 | 1.38 | 0.49 | 0.26 | 0.1 | 0.04 | 0.02 |
| | $\sigma = 0.25$ | K=500 | 68.88 | 63.53 | 47.56 | 22.19 | 5.8 | 1.54 | 0.53 | 0.26 | 0.1 | 0.06 | 0.03 |
| | $\sigma = 0.25$ | K=600 | 68.09 | 63.21 | 47.78 | 22.05 | 6.16 | 1.59 | 0.51 | 0.25 | 0.12 | 0.07 | 0.02 |
| | $\sigma = 0.25$ | K=700 | 67.57 | 63.02 | 47.6 | 22.25 | 5.97 | 1.63 | 0.57 | 0.29 | 0.11 | 0.04 | 0.02 |
| | $\sigma = 0.25$ | K=800 | 67.36 | 62.93 | 47.64 | 22.04 | 6.29 | 1.62 | 0.6 | 0.27 | 0.11 | 0.04 | 0.02 |
| | $\sigma = 0.25$ | K=900 | 67.22 | 62.93 | 47.45 | 22.55 | 6.19 | 1.62 | 0.54 | 0.26 | 0.11 | 0.05 | 0.03 |
| | $\sigma = 0.50$ | K=100 | 63.18 | 55.88 | 47.07 | 37.2 | 26.56 | 16.43 | 8.0 | 3.21 | 1.23 | 0.55 | 0.19 |
| | $\sigma = 0.50$ | K=200 | 61.26 | 55.08 | 47.25 | 37.86 | 27.25 | 16.49 | 7.49 | 2.56 | 1.07 | 0.53 | 0.23 |
| | $\sigma = 0.50$ | K=300 | 59.52 | 54.25 | 47.35 | 38.28 | 27.29 | 16.23 | 7.16 | 2.39 | 0.96 | 0.48 | 0.24 |
| | $\sigma = 0.50$ | K=400 | 58.29 | 53.67 | 47.19 | 38.05 | 27.45 | 16.39 | 7.41 | 2.44 | 0.93 | 0.45 | 0.24 |
| | $\sigma = 0.50$ | K=500 | 57.46 | 53.53 | 47.38 | 38.28 | 27.47 | 16.45 | 7.38 | 2.38 | 0.87 | 0.48 | 0.24 |
| | $\sigma = 0.50$ | K=600 | 56.68 | 53.11 | 47.04 | 38.21 | 27.47 | 16.34 | 7.21 | 2.37 | 1.03 | 0.55 | 0.32 |
| | $\sigma = 0.50$ | K=700 | 55.83 | 52.37 | 46.88 | 38.12 | 27.43 | 16.37 | 7.21 | 2.3 | 1.01 | 0.57 | 0.37 |
| | $\sigma = 0.50$ | K=800 | 55.26 | 52.11 | 46.8 | 38.3 | 27.26 | 16.19 | 7.18 | 2.45 | 1.04 | 0.62 | 0.4 |
| | $\sigma = 0.50$ | K=900 | 54.83 | 51.83 | 46.62 | 38.15 | 27.55 | 16.5 | 7.37 | 2.52 | 1.21 | 0.69 | 0.5 |

Table 5: **Certified accuracy per radius on CIFAR10.** We report Cohen-DS$^2$ under varying $\sigma$ and number of iterations $K$ in Algorithm 1.

| | $\ell_2^r$ (CIFAR10) | | 0.0 | 0.25 | 0.50 | 0.75 | 1.00 | 1.25 | 1.50 | 1.75 | 2.00 | 2.25 | 2.50 |
|---|---|---|---|---|---|---|---|---|---|---|---|---|---|
| Cohen-DS$^2$ | $\sigma$ =0.12 | K=100 | 79.8 | 60.56 | 26.87 | 0.0 | 0.0 | 0.0 | 0.0 | 0.0 | 0.0 | 0.0 | 0.0 |
| | $\sigma$ =0.12 | K=100 | 79.8 | 60.56 | 26.87 | 0.0 | 0.0 | 0.0 | 0.0 | 0.0 | 0.0 | 0.0 | 0.0 |
| | $\sigma$ =0.12 | K=200 | 79.83 | 62.05 | 28.13 | 0.0 | 0.0 | 0.0 | 0.0 | 0.0 | 0.0 | 0.0 | 0.0 |
| | $\sigma$ =0.12 | K=300 | 79.74 | 62.81 | 26.96 | 0.03 | 0.0 | 0.0 | 0.0 | 0.0 | 0.0 | 0.0 | 0.0 |
| | $\sigma$ =0.12 | K=400 | 79.56 | 63.07 | 25.47 | 6.66 | 0.0 | 0.0 | 0.0 | 0.0 | 0.0 | 0.0 | 0.0 |
| | $\sigma$ =0.12 | K=500 | 79.4 | 63.24 | 24.23 | 7.74 | 0.0 | 0.0 | 0.0 | 0.0 | 0.0 | 0.0 | 0.0 |
| | $\sigma$ =0.12 | K=600 | 79.14 | 63.23 | 23.58 | 7.5 | 0.0 | 0.0 | 0.0 | 0.0 | 0.0 | 0.0 | 0.0 |
| | $\sigma$ =0.12 | K=700 | 78.95 | 63.34 | 22.96 | 7.12 | 0.86 | 0.0 | 0.0 | 0.0 | 0.0 | 0.0 | 0.0 |
| | $\sigma$ =0.12 | K=800 | 78.77 | 63.34 | 22.57 | 6.48 | 1.26 | 0.0 | 0.0 | 0.0 | 0.0 | 0.0 | 0.0 |
| | $\sigma$ =0.12 | K=900 | 79.06 | 64.6 | 22.69 | 6.61 | 1.68 | 0.01 | 0.0 | 0.0 | 0.0 | 0.0 | 0.0 |
| | $\sigma$ =0.12 | K=1000 | 79.02 | 64.54 | 22.27 | 6.27 | 1.69 | 0.02 | 0.0 | 0.0 | 0.0 | 0.0 | 0.0 |
| | $\sigma$ =0.12 | K=1100 | 78.81 | 64.41 | 21.9 | 5.89 | 1.58 | 0.22 | 0.0 | 0.0 | 0.0 | 0.0 | 0.0 |
| | $\sigma$ =0.12 | K=1200 | 78.7 | 64.37 | 21.88 | 5.45 | 1.42 | 0.27 | 0.0 | 0.0 | 0.0 | 0.0 | 0.0 |
| | $\sigma$ =0.12 | K=1300 | 78.53 | 64.39 | 21.67 | 5.15 | 1.31 | 0.26 | 0.0 | 0.0 | 0.0 | 0.0 | 0.0 |
| | $\sigma$ =0.12 | K=1400 | 78.39 | 64.46 | 21.55 | 4.96 | 1.13 | 0.29 | 0.01 | 0.0 | 0.0 | 0.0 | 0.0 |
| | $\sigma$ =0.12 | K=1500 | 78.31 | 64.41 | 21.56 | 4.73 | 1.05 | 0.3 | 0.03 | 0.0 | 0.0 | 0.0 | 0.0 |
| | $\sigma$ =0.25 | K=100 | 74.99 | 61.47 | 43.92 | 24.54 | 7.93 | 0.0 | 0.0 | 0.0 | 0.0 | 0.0 | 0.0 |
| | $\sigma$ =0.25 | K=200 | 75.13 | 63.21 | 45.94 | 25.75 | 9.35 | 0.0 | 0.0 | 0.0 | 0.0 | 0.0 | 0.0 |
| | $\sigma$ =0.25 | K=300 | 75.0 | 64.03 | 46.96 | 25.96 | 10.05 | 0.0 | 0.0 | 0.0 | 0.0 | 0.0 | 0.0 |
| | $\sigma$ =0.25 | K=400 | 75.04 | 64.58 | 47.59 | 25.63 | 9.93 | 1.92 | 0.0 | 0.0 | 0.0 | 0.0 | 0.0 |
| | $\sigma$ =0.25 | K=500 | 74.79 | 64.9 | 47.85 | 25.41 | 9.42 | 2.6 | 0.0 | 0.0 | 0.0 | 0.0 | 0.0 |
| | $\sigma$ =0.25 | K=600 | 74.7 | 65.15 | 48.38 | 25.05 | 8.88 | 2.69 | 0.0 | 0.0 | 0.0 | 0.0 | 0.0 |
| | $\sigma$ =0.25 | K=700 | 74.51 | 65.35 | 48.47 | 24.71 | 8.34 | 2.62 | 0.01 | 0.0 | 0.0 | 0.0 | 0.0 |
| | $\sigma$ =0.25 | K=800 | 74.46 | 65.42 | 48.5 | 24.72 | 7.98 | 2.43 | 0.52 | 0.0 | 0.0 | 0.0 | 0.0 |
| | $\sigma$ =0.25 | K=900 | 74.58 | 66.42 | 50.23 | 25.57 | 8.25 | 2.83 | 0.74 | 0.0 | 0.0 | 0.0 | 0.0 |
| | $\sigma$ =0.25 | K=1000 | 74.39 | 66.47 | 50.17 | 25.41 | 7.9 | 2.63 | 0.75 | 0.01 | 0.0 | 0.0 | 0.0 |
| | $\sigma$ =0.25 | K=1100 | 74.2 | 66.42 | 50.31 | 25.13 | 7.65 | 2.41 | 0.72 | 0.14 | 0.0 | 0.0 | 0.0 |
| | $\sigma$ =0.25 | K=1200 | 74.12 | 66.37 | 50.37 | 24.92 | 7.36 | 2.31 | 0.65 | 0.18 | 0.0 | 0.0 | 0.0 |
| | $\sigma$ =0.25 | K=1300 | 73.98 | 66.41 | 50.38 | 24.75 | 7.14 | 2.25 | 0.58 | 0.19 | 0.0 | 0.0 | 0.0 |
| | $\sigma$ =0.25 | K=1400 | 73.81 | 66.39 | 50.41 | 24.79 | 6.85 | 2.2 | 0.51 | 0.15 | 0.03 | 0.0 | 0.0 |
| | $\sigma$ =0.25 | K=1500 | 73.67 | 66.33 | 50.31 | 24.63 | 6.68 | 2.03 | 0.54 | 0.19 | 0.03 | 0.0 | 0.0 |
| | $\sigma$ =0.50 | K=100 | 63.92 | 53.49 | 42.6 | 31.83 | 22.15 | 14.12 | 7.48 | 3.51 | 0.0 | 0.0 | 0.0 |
| | $\sigma$ =0.50 | K=200 | 64.14 | 54.36 | 44.3 | 33.52 | 23.79 | 15.14 | 7.93 | 3.6 | 1.09 | 0.0 | 0.0 |
| | $\sigma$ =0.50 | K=300 | 64.21 | 54.95 | 45.32 | 35.04 | 24.71 | 15.81 | 8.16 | 3.65 | 1.36 | 0.0 | 0.0 |
| | $\sigma$ =0.50 | K=400 | 64.22 | 55.56 | 45.92 | 35.86 | 25.45 | 16.28 | 8.35 | 3.79 | 1.41 | 0.0 | 0.0 |
| | $\sigma$ =0.50 | K=500 | 64.14 | 55.84 | 46.35 | 36.29 | 25.88 | 16.56 | 8.39 | 3.69 | 1.42 | 0.3 | 0.0 |
| | $\sigma$ =0.50 | K=600 | 64.14 | 56.07 | 46.7 | 36.71 | 26.18 | 16.76 | 8.48 | 3.61 | 1.41 | 0.45 | 0.0 |
| | $\sigma$ =0.50 | K=700 | 64.04 | 56.2 | 46.94 | 37.09 | 26.54 | 16.76 | 8.4 | 3.63 | 1.37 | 0.45 | 0.0 |
| | $\sigma$ =0.50 | K=800 | 63.93 | 56.32 | 47.23 | 37.33 | 26.69 | 16.91 | 8.35 | 3.46 | 1.28 | 0.5 | 0.09 |
| | $\sigma$ =0.50 | K=900 | 64.26 | 57.26 | 48.27 | 38.85 | 28.41 | 17.97 | 8.82 | 3.66 | 1.37 | 0.58 | 0.14 |
| | $\sigma$ =0.50 | K=1000 | 64.06 | 57.26 | 48.41 | 38.96 | 28.49 | 18.1 | 8.65 | 3.64 | 1.33 | 0.6 | 0.21 |
| | $\sigma$ =0.50 | K=1100 | 63.72 | 57.21 | 48.47 | 39.0 | 28.69 | 18.26 | 8.57 | 3.55 | 1.36 | 0.59 | 0.2 |
| | $\sigma$ =0.50 | K=1200 | 63.56 | 57.15 | 48.67 | 38.96 | 28.81 | 18.18 | 8.53 | 3.52 | 1.32 | 0.58 | 0.23 |
| | $\sigma$ =0.50 | K=1300 | 63.29 | 57.01 | 48.81 | 39.07 | 28.98 | 18.31 | 8.44 | 3.44 | 1.3 | 0.6 | 0.21 |
| | $\sigma$ =0.50 | K=1400 | 63.09 | 56.9 | 48.88 | 39.11 | 29.07 | 18.22 | 8.62 | 3.3 | 1.34 | 0.56 | 0.22 |
| | $\sigma$ =0.50 | K=1500 | 62.94 | 56.87 | 48.9 | 39.21 | 29.04 | 18.1 | 8.55 | 3.27 | 1.28 | 0.53 | 0.23 |

Table 6: **Certified accuracy per radius on ImageNet.** We compare Cohen against Cohen-DS and Cohen-DS$^2$ under varying $\sigma$ and number of iterations $K$ in Algorithm 1.

| | $\ell_2^r$ (ImageNet) | | 0.0 | 0.25 | 0.50 | 0.75 | 1.00 | 1.50 | 2.0 | 2.5 | 3.0 | 3.50 | 4.0 |
|---|---|---|---|---|---|---|---|---|---|---|---|---|---|
| **Cohen** | $\sigma = 0.25$ | | 66.6 | 58.2 | 49.0 | 38.0 | 0.0 | 0.0 | 0.0 | 0.0 | 0.0 | 0.0 | 0.0 |
| | $\sigma = 0.50$ | | 57.2 | 51.4 | 45.8 | 42.4 | 37.4 | 27.8 | 0.0 | 0.0 | 0.0 | 0.0 | 0.0 |
| | $\sigma = 1.0$ | | 43.6 | 40.6 | 37.8 | 35.4 | 32.6 | 25.8 | 19.4 | 14.4 | 12.0 | 8.6 | 0.0 |
| **Cohen-DS** | $\sigma =0.25$ | K=100 | 67.8 | 61.0 | 53.6 | 42.8 | 18.8 | 0.0 | 0.0 | 0.0 | 0.0 | 0.0 | 0.0 |
| | $\sigma =0.25$ | K=200 | 67.0 | 61.4 | 53.6 | 43.0 | 18.0 | 0.0 | 0.0 | 0.0 | 0.0 | 0.0 | 0.0 |
| | $\sigma =0.25$ | K=300 | 66.8 | 61.2 | 53.4 | 42.2 | 18.6 | 0.0 | 0.0 | 0.0 | 0.0 | 0.0 | 0.0 |
| | $\sigma =0.25$ | K=400 | 66.2 | 61.4 | 53.2 | 42.2 | 18.0 | 0.0 | 0.0 | 0.0 | 0.0 | 0.0 | 0.0 |
| | $\sigma =0.50$ | K=100 | 58.4 | 54.0 | 48.2 | 45.2 | 40.6 | 30.4 | 1.8 | 0.0 | 0.0 | 0.0 | 0.0 |
| | $\sigma =0.50$ | K=200 | 58.0 | 53.4 | 48.2 | 45.2 | 41.4 | 29.8 | 9.0 | 0.0 | 0.0 | 0.0 | 0.0 |
| | $\sigma =0.50$ | K=300 | 58.0 | 54.0 | 48.8 | 45.4 | 41.4 | 30.2 | 9.0 | 0.0 | 0.0 | 0.0 | 0.0 |
| | $\sigma =0.50$ | K=400 | 57.8 | 53.8 | 48.8 | 45.6 | 42.0 | 30.4 | 8.2 | 0.0 | 0.0 | 0.0 | 0.0 |
| | $\sigma =1.0$ | K=100 | 45.0 | 42.6 | 40.4 | 39.0 | 36.4 | 29.6 | 22.4 | 17.8 | 13.8 | 10.0 | 0.2 |
| | $\sigma =1.0$ | K=200 | 45.2 | 43.0 | 41.8 | 39.4 | 36.8 | 29.6 | 23.0 | 18.6 | 14.2 | 10.2 | 0.6 |
| | $\sigma =1.0$ | K=300 | 45.0 | 43.4 | 41.2 | 39.6 | 37.2 | 30.0 | 23.4 | 18.8 | 14.4 | 9.4 | 2.0 |
| | $\sigma =1.0$ | K=400 | 44.8 | 43.2 | 41.4 | 39.6 | 37.2 | 30.4 | 23.2 | 18.8 | 14.6 | 9.8 | 1.8 |
| **Cohen-DS$^2$** | $\sigma =0.25$ | K=100 | 67.2 | 64.2 | 58.4 | 45.4 | 17.8 | 0.0 | 0.0 | 0.0 | 0.0 | 0.0 | 0.0 |
| | $\sigma =0.25$ | K=200 | 66.8 | 64.2 | 58.2 | 45.6 | 18.0 | 0.0 | 0.0 | 0.0 | 0.0 | 0.0 | 0.0 |
| | $\sigma =0.25$ | K=300 | 66.6 | 64.2 | 58.0 | 45.2 | 18.4 | 0.0 | 0.0 | 0.0 | 0.0 | 0.0 | 0.0 |
| | $\sigma =0.25$ | K=400 | 67.4 | 64.2 | 58.2 | 45.0 | 18.0 | 0.0 | 0.0 | 0.0 | 0.0 | 0.0 | 0.0 |
| | $\sigma =0.50$ | K=100 | 58.0 | 55.2 | 51.6 | 46.2 | 41.2 | 30.2 | 2.2 | 0.0 | 0.0 | 0.0 | 0.0 |
| | $\sigma =0.50$ | K=200 | 57.6 | 55.2 | 51.8 | 47.0 | 41.8 | 30.4 | 8.0 | 0.0 | 0.0 | 0.0 | 0.0 |
| | $\sigma =0.50$ | K=300 | 57.6 | 55.0 | 51.8 | 46.8 | 41.8 | 30.4 | 8.0 | 0.0 | 0.0 | 0.0 | 0.0 |
| | $\sigma =0.50$ | K=400 | 57.4 | 55.4 | 51.6 | 47.4 | 41.8 | 30.6 | 8.2 | 0.0 | 0.0 | 0.0 | 0.0 |
| | $\sigma =1.0$ | K=100 | 46.4 | 44.6 | 41.4 | 38.6 | 37.2 | 31.4 | 24.8 | 20.6 | 16.6 | 11.0 | 0.4 |
| | $\sigma =1.0$ | K=200 | 46.6 | 44.4 | 42.0 | 39.2 | 37.6 | 31.2 | 25.0 | 20.8 | 17.0 | 10.8 | 0.4 |
| | $\sigma =1.0$ | K=300 | 46.0 | 44.6 | 41.8 | 39.2 | 37.4 | 31.4 | 24.6 | 20.8 | 17.2 | 11.0 | 1.8 |
| | $\sigma =1.0$ | K=400 | 46.8 | 45.0 | 42.6 | 39.4 | 37.6 | 31.8 | 24.8 | 21.2 | 16.8 | 11.0 | 2.0 |

Table 7: **Certified accuracy per radius on CIFAR10.** We compare SmoothAdv against SmoothAdv-DS and SmoothAdv-DS$^2$ under varying $\sigma$ and number of iterations $K$ in Algorithm 1.

| | $\ell_2^r$ (CIFAR10) | | 0.0 | 0.25 | 0.50 | 0.75 | 1.00 | 1.25 | 1.50 | 1.75 | 2.00 | 2.25 | 2.50 |
|---|---|---|---|---|---|---|---|---|---|---|---|---|---|
| **SmoothAdv** | $\sigma = 0.12$ | | 75.97 | 62.44 | 0.0 | 0.0 | 0.0 | 0.0 | 0.0 | 0.0 | 0.0 | 0.0 | 0.0 |
| | $\sigma = 0.25$ | | 70.82 | 59.55 | 46.71 | 33.66 | 0.0 | 0.0 | 0.0 | 0.0 | 0.0 | 0.0 | 0.0 |
| | $\sigma = 0.50$ | | 60.96 | 52.6 | 43.5 | 34.62 | 26.53 | 19.49 | 12.9 | 7.47 | 0.0 | 0.0 | 0.0 |
| **SmoothAdv-DS** | $\sigma = 0.12$ | K=100 | 75.74 | 63.58 | 40.88 | 0.0 | 0.0 | 0.0 | 0.0 | 0.0 | 0.0 | 0.0 | 0.0 |
| | $\sigma = 0.12$ | K=200 | 75.7 | 64.39 | 45.05 | 0.0 | 0.0 | 0.0 | 0.0 | 0.0 | 0.0 | 0.0 | 0.0 |
| | $\sigma = 0.12$ | K=300 | 75.69 | 64.97 | 46.13 | 0.55 | 0.0 | 0.0 | 0.0 | 0.0 | 0.0 | 0.0 | 0.0 |
| | $\sigma = 0.12$ | K=400 | 75.73 | 65.43 | 46.39 | 22.49 | 0.0 | 0.0 | 0.0 | 0.0 | 0.0 | 0.0 | 0.0 |
| | $\sigma = 0.12$ | K=500 | 75.74 | 65.75 | 46.57 | 25.06 | 0.03 | 0.0 | 0.0 | 0.0 | 0.0 | 0.0 | 0.0 |
| | $\sigma = 0.12$ | K=600 | 75.72 | 66.04 | 46.64 | 25.16 | 0.24 | 0.0 | 0.0 | 0.0 | 0.0 | 0.0 | 0.0 |
| | $\sigma = 0.12$ | K=700 | 75.66 | 66.23 | 46.74 | 24.64 | 7.26 | 0.0 | 0.0 | 0.0 | 0.0 | 0.0 | 0.0 |
| | $\sigma = 0.12$ | K=800 | 75.65 | 66.3 | 46.61 | 23.97 | 11.54 | 0.0 | 0.0 | 0.0 | 0.0 | 0.0 | 0.0 |
| | $\sigma = 0.12$ | K=900 | 75.64 | 66.44 | 46.43 | 23.48 | 11.75 | 0.03 | 0.0 | 0.0 | 0.0 | 0.0 | 0.0 |
| | $\sigma = 0.25$ | K=100 | 71.34 | 60.81 | 48.38 | 35.14 | 17.76 | 0.0 | 0.0 | 0.0 | 0.0 | 0.0 | 0.0 |
| | $\sigma = 0.25$ | K=200 | 71.32 | 61.38 | 49.44 | 36.24 | 20.71 | 0.01 | 0.0 | 0.0 | 0.0 | 0.0 | 0.0 |
| | $\sigma = 0.25$ | K=300 | 71.3 | 62.01 | 50.16 | 36.9 | 21.69 | 0.28 | 0.0 | 0.0 | 0.0 | 0.0 | 0.0 |
| | $\sigma = 0.25$ | K=400 | 71.32 | 62.45 | 50.76 | 37.24 | 22.33 | 8.37 | 0.01 | 0.0 | 0.0 | 0.0 | 0.0 |
| | $\sigma = 0.25$ | K=500 | 71.23 | 62.82 | 51.27 | 37.46 | 22.42 | 10.67 | 0.01 | 0.0 | 0.0 | 0.0 | 0.0 |
| | $\sigma = 0.25$ | K=600 | 71.26 | 63.02 | 51.66 | 37.66 | 22.04 | 11.06 | 0.06 | 0.0 | 0.0 | 0.0 | 0.0 |
| | $\sigma = 0.25$ | K=700 | 71.12 | 63.26 | 51.72 | 37.61 | 21.82 | 11.0 | 0.52 | 0.0 | 0.0 | 0.0 | 0.0 |
| | $\sigma = 0.25$ | K=800 | 71.07 | 63.4 | 51.94 | 37.5 | 21.43 | 10.6 | 4.26 | 0.0 | 0.0 | 0.0 | 0.0 |
| | $\sigma = 0.25$ | K=900 | 71.04 | 63.54 | 52.1 | 37.38 | 21.18 | 10.1 | 4.63 | 0.0 | 0.0 | 0.0 | 0.0 |
| | $\sigma = 0.50$ | K=100 | 61.16 | 53.06 | 44.28 | 35.42 | 27.29 | 20.18 | 13.6 | 7.82 | 0.02 | 0.0 | 0.0 |
| | $\sigma = 0.50$ | K=200 | 61.22 | 53.44 | 44.96 | 36.11 | 28.0 | 20.81 | 13.98 | 8.25 | 2.85 | 0.0 | 0.0 |
| | $\sigma = 0.50$ | K=300 | 61.24 | 53.74 | 45.39 | 36.81 | 28.72 | 21.21 | 14.23 | 8.29 | 3.29 | 0.0 | 0.0 |
| | $\sigma = 0.50$ | K=400 | 61.22 | 53.95 | 45.65 | 37.29 | 29.22 | 21.58 | 14.53 | 8.42 | 3.78 | 0.05 | 0.0 |
| | $\sigma = 0.50$ | K=500 | 61.21 | 54.15 | 46.03 | 37.8 | 29.54 | 21.72 | 14.73 | 8.43 | 3.99 | 1.02 | 0.0 |
| | $\sigma = 0.50$ | K=600 | 61.2 | 54.3 | 46.42 | 38.11 | 29.83 | 21.94 | 14.95 | 8.42 | 4.07 | 1.57 | 0.0 |
| | $\sigma = 0.50$ | K=700 | 61.23 | 54.47 | 46.58 | 38.39 | 30.24 | 22.04 | 14.95 | 8.5 | 4.12 | 1.77 | 0.03 |
| | $\sigma = 0.50$ | K=800 | 61.19 | 54.58 | 46.73 | 38.65 | 30.39 | 22.15 | 14.86 | 8.49 | 4.09 | 1.83 | 0.43 |
| | $\sigma = 0.50$ | K=900 | 61.25 | 54.65 | 46.88 | 38.82 | 30.6 | 22.19 | 14.89 | 8.49 | 4.17 | 1.84 | 0.6 |
| **SmoothAdv-DS$^2$** | $\sigma = 0.12$ | K=100 | 76.04 | 63.62 | 41.88 | 0.0 | 0.0 | 0.0 | 0.0 | 0.0 | 0.0 | 0.0 | 0.0 |
| | $\sigma = 0.12$ | K=200 | 76.03 | 64.54 | 46.4 | 0.01 | 0.0 | 0.0 | 0.0 | 0.0 | 0.0 | 0.0 | 0.0 |
| | $\sigma = 0.12$ | K=300 | 76.0 | 65.36 | 47.36 | 0.82 | 0.0 | 0.0 | 0.0 | 0.0 | 0.0 | 0.0 | 0.0 |
| | $\sigma = 0.12$ | K=400 | 75.99 | 65.85 | 47.98 | 23.18 | 0.0 | 0.0 | 0.0 | 0.0 | 0.0 | 0.0 | 0.0 |
| | $\sigma = 0.12$ | K=500 | 76.11 | 66.15 | 48.16 | 26.09 | 0.07 | 0.0 | 0.0 | 0.0 | 0.0 | 0.0 | 0.0 |
| | $\sigma = 0.12$ | K=600 | 76.14 | 66.39 | 48.13 | 26.08 | 0.47 | 0.0 | 0.0 | 0.0 | 0.0 | 0.0 | 0.0 |
| | $\sigma = 0.12$ | K=700 | 76.15 | 66.52 | 48.1 | 25.73 | 7.99 | 0.0 | 0.0 | 0.0 | 0.0 | 0.0 | 0.0 |
| | $\sigma = 0.12$ | K=800 | 76.15 | 66.69 | 47.84 | 25.16 | 11.89 | 0.02 | 0.0 | 0.0 | 0.0 | 0.0 | 0.0 |
| | $\sigma = 0.12$ | K=900 | 76.05 | 66.77 | 47.9 | 24.34 | 11.82 | 0.06 | 0.0 | 0.0 | 0.0 | 0.0 | 0.0 |
| | $\sigma = 0.25$ | K=100 | 71.2 | 60.56 | 48.36 | 35.19 | 17.76 | 0.0 | 0.0 | 0.0 | 0.0 | 0.0 | 0.0 |
| | $\sigma = 0.25$ | K=200 | 71.27 | 61.55 | 49.57 | 36.45 | 21.31 | 0.01 | 0.0 | 0.0 | 0.0 | 0.0 | 0.0 |
| | $\sigma = 0.25$ | K=300 | 71.3 | 62.16 | 50.75 | 37.41 | 22.66 | 0.48 | 0.0 | 0.0 | 0.0 | 0.0 | 0.0 |
| | $\sigma = 0.25$ | K=400 | 71.41 | 62.76 | 51.44 | 37.85 | 23.18 | 9.12 | 0.01 | 0.0 | 0.0 | 0.0 | 0.0 |
| | $\sigma = 0.25$ | K=500 | 71.37 | 63.0 | 51.89 | 38.06 | 23.16 | 11.37 | 0.04 | 0.0 | 0.0 | 0.0 | 0.0 |
| | $\sigma = 0.25$ | K=600 | 71.37 | 63.36 | 52.25 | 38.31 | 22.8 | 11.84 | 0.16 | 0.0 | 0.0 | 0.0 | 0.0 |
| | $\sigma = 0.25$ | K=700 | 71.35 | 63.45 | 52.43 | 38.33 | 22.58 | 11.61 | 0.73 | 0.0 | 0.0 | 0.0 | 0.0 |
| | $\sigma = 0.25$ | K=800 | 71.25 | 63.65 | 52.67 | 38.26 | 22.35 | 11.17 | 4.69 | 0.0 | 0.0 | 0.0 | 0.0 |
| | $\sigma = 0.25$ | K=900 | 71.21 | 63.85 | 52.81 | 38.21 | 22.21 | 10.75 | 4.92 | 0.03 | 0.0 | 0.0 | 0.0 |
| | $\sigma = 0.50$ | K=100 | 61.08 | 53.0 | 44.33 | 35.59 | 27.49 | 20.09 | 13.74 | 7.98 | 0.04 | 0.0 | 0.0 |
| | $\sigma = 0.50$ | K=200 | 61.07 | 53.41 | 44.95 | 36.33 | 28.39 | 20.86 | 14.05 | 8.22 | 2.9 | 0.0 | 0.0 |
| | $\sigma = 0.50$ | K=300 | 61.1 | 53.8 | 45.61 | 37.13 | 28.9 | 21.5 | 14.32 | 8.56 | 3.57 | 0.0 | 0.0 |
| | $\sigma = 0.50$ | K=400 | 61.1 | 54.14 | 46.02 | 37.77 | 29.3 | 21.94 | 14.66 | 8.63 | 3.89 | 0.06 | 0.0 |
| | $\sigma = 0.50$ | K=500 | 61.15 | 54.21 | 46.52 | 38.15 | 29.79 | 22.21 | 14.91 | 8.66 | 4.23 | 1.05 | 0.0 |
| | $\sigma = 0.50$ | K=600 | 61.2 | 54.33 | 46.89 | 38.59 | 30.08 | 22.35 | 15.01 | 8.74 | 4.28 | 1.56 | 0.01 |
| | $\sigma = 0.50$ | K=700 | 61.18 | 54.56 | 47.11 | 38.93 | 30.4 | 22.51 | 15.12 | 8.85 | 4.34 | 1.77 | 0.03 |
| | $\sigma = 0.50$ | K=800 | 61.15 | 54.72 | 47.41 | 39.17 | 30.59 | 22.56 | 15.14 | 8.75 | 4.31 | 1.85 | 0.53 |
| | $\sigma = 0.50$ | K=900 | 61.12 | 54.78 | 47.62 | 39.32 | 30.78 | 22.64 | 15.14 | 8.73 | 4.26 | 1.94 | 0.71 |

Table 8: **Certified accuracy per radius on ImageNet.** We compare SmoothAdv against SmoothAdv-DS and SmoothAdv-DS$^2$ under varying $\sigma$ and number of iterations $K$ in Algorithm 1.

| | $\ell_2^r$ (ImageNet) | | 0.0 | 0.25 | 0.50 | 0.75 | 1.00 | 1.50 | 2.0 | 2.5 | 3.0 | 3.50 | 4.0 |
|---|---|---|---|---|---|---|---|---|---|---|---|---|---|
| **SmoothAdv** | $\sigma = 0.25$ | | 60.8 | 57.8 | 54.6 | 50.4 | 0.0 | 0.0 | 0.0 | 0.0 | 0.0 | 0.0 | 0.0 |
| | $\sigma = 0.50$ | | 54.6 | 52.6 | 48.8 | 44.6 | 42.2 | 35.6 | 0.0 | 0.0 | 0.0 | 0.0 | 0.0 |
| | $\sigma = 1.0$ | | 40.6 | 39.6 | 38.6 | 36.4 | 33.6 | 29.8 | 25.6 | 20.4 | 18.0 | 14.2 | 0.0 |
| **SmoothAdv-DS** | $\sigma =0.25$ | K=100 | 61.6 | 59.6 | 56.8 | 52.6 | 31.4 | 0.0 | 0.0 | 0.0 | 0.0 | 0.0 | 0.0 |
| | $\sigma =0.25$ | K=200 | 61.6 | 59.8 | 57.2 | 52.8 | 35.8 | 0.0 | 0.0 | 0.0 | 0.0 | 0.0 | 0.0 |
| | $\sigma =0.25$ | K=300 | 62.0 | 60.2 | 57.2 | 52.8 | 36.6 | 0.0 | 0.0 | 0.0 | 0.0 | 0.0 | 0.0 |
| | $\sigma =0.25$ | K=400 | 61.8 | 60.4 | 57.4 | 53.2 | 36.8 | 0.0 | 0.0 | 0.0 | 0.0 | 0.0 | 0.0 |
| | $\sigma =0.50$ | K=100 | 55.0 | 53.6 | 51.2 | 47.2 | 45.2 | 38.0 | 4.8 | 0.0 | 0.0 | 0.0 | 0.0 |
| | $\sigma =0.50$ | K=200 | 55.0 | 53.8 | 51.6 | 48.4 | 46.4 | 39.2 | 16.6 | 0.0 | 0.0 | 0.0 | 0.0 |
| | $\sigma =0.50$ | K=300 | 55.4 | 54.0 | 51.6 | 48.6 | 47.0 | 39.2 | 18.0 | 0.0 | 0.0 | 0.0 | 0.0 |
| | $\sigma =0.50$ | K=400 | 55.2 | 54.0 | 51.6 | 48.8 | 47.0 | 39.0 | 18.6 | 0.0 | 0.0 | 0.0 | 0.0 |
| | $\sigma =1.0$ | K=100 | 41.8 | 41.0 | 39.4 | 37.6 | 35.2 | 31.6 | 28.0 | 22.6 | 19.2 | 15.2 | 0.8 |
| | $\sigma =1.0$ | K=200 | 42.4 | 41.8 | 40.2 | 38.4 | 36.6 | 32.4 | 28.8 | 23.4 | 19.0 | 14.6 | 1.2 |
| | $\sigma =1.0$ | K=300 | 42.6 | 41.8 | 40.4 | 38.8 | 36.8 | 32.4 | 29.2 | 23.8 | 19.6 | 15.2 | 6.2 |
| | $\sigma =1.0$ | K=400 | 42.8 | 42.2 | 40.8 | 38.8 | 37.0 | 33.2 | 29.0 | 23.8 | 19.6 | 14.8 | 6.2 |
| **SmoothAdv-DS$^2$** | $\sigma =0.25$ | K=100 | 62.2 | 60.4 | 58.8 | 54.0 | 27.0 | 0.0 | 0.0 | 0.0 | 0.0 | 0.0 | 0.0 |
| | $\sigma =0.25$ | K=200 | 62.0 | 60.6 | 58.6 | 54.2 | 27.4 | 0.0 | 0.0 | 0.0 | 0.0 | 0.0 | 0.0 |
| | $\sigma =0.25$ | K=300 | 62.0 | 60.4 | 58.8 | 54.0 | 27.4 | 0.0 | 0.0 | 0.0 | 0.0 | 0.0 | 0.0 |
| | $\sigma =0.25$ | K=400 | 61.8 | 60.4 | 58.8 | 54.0 | 27.4 | 0.0 | 0.0 | 0.0 | 0.0 | 0.0 | 0.0 |
| | $\sigma =0.50$ | K=100 | 55.8 | 54.2 | 52.6 | 50.4 | 48.2 | 43.0 | 7.8 | 0.0 | 0.0 | 0.0 | 0.0 |
| | $\sigma =0.50$ | K=200 | 55.2 | 54.0 | 51.8 | 49.8 | 47.8 | 42.6 | 14.2 | 0.0 | 0.0 | 0.0 | 0.0 |
| | $\sigma =0.50$ | K=300 | 55.6 | 54.0 | 52.0 | 49.8 | 47.8 | 42.6 | 15.0 | 0.0 | 0.0 | 0.0 | 0.0 |
| | $\sigma =0.50$ | K=400 | 55.6 | 54.4 | 52.2 | 50.2 | 48.2 | 43.0 | 15.0 | 0.0 | 0.0 | 0.0 | 0.0 |
| | $\sigma =1.0$ | K=100 | 44.0 | 43.0 | 41.2 | 40.6 | 38.4 | 34.6 | 30.6 | 25.4 | 21.6 | 18.6 | 1.2 |
| | $\sigma =1.0$ | K=200 | 44.4 | 43.2 | 41.6 | 40.6 | 38.6 | 34.8 | 30.6 | 25.0 | 21.6 | 18.4 | 1.6 |
| | $\sigma =1.0$ | K=300 | 44.2 | 43.0 | 41.8 | 41.2 | 38.6 | 34.6 | 30.6 | 25.2 | 21.4 | 17.8 | 4.2 |
| | $\sigma =1.0$ | K=400 | 43.8 | 43.0 | 41.0 | 40.8 | 38.6 | 34.6 | 30.2 | 25.2 | 21.4 | 18.2 | 4.0 |

### F.3    MACER VS MACER-DS VS MACER-DS$^2$ (N=1) VS MACER-DS$^2$ (N=8)

We report $\ell_2^r$ certified accuracy per radius $r$ for MACER (Zhai et al., 2020) variants on CIFAR10. Note that as highlighted in the main manuscript, for certification only, *i.e.* $MACER - DS$, we set $n = 8$ for all experiments in Algorithm 1. Moreover, in the main paper and for ease of computation we set $n = 1$ for when training is employed, *i.e.* $-DS^2$. In here we also explore the variant where when data dependent smoothing is introduced during training we set $n = 8$ for ablations. We refer to when $n = 1$ and $n = 8$ for when data dependent smoothing is used in training and certification as $MACER - DS(n = 1)$ and $MACER - DS(n = 8)$, respectively.

Table 9: **Certified accuracy per radius on CIFAR10.** We compare MACER against MACER-DS and MACER-DS$^2(n=1)$ under varying $\sigma$ and number of iterations $K$ in Algorithm 1.

| | $\ell_2^r$ (CIFAR10) | 0.0 | 0.25 | 0.50 | 0.75 | 1.00 | 1.25 | 1.50 | 1.75 | 2.00 | 2.25 | 2.50 |
|---|---|---|---|---|---|---|---|---|---|---|---|---|
| **MACER** | $\sigma = 0.12$ | 78.75 | 58.51 | 0.0 | 0.0 | 0.0 | 0.0 | 0.0 | 0.0 | 0.0 | 0.0 | 0.0 |
| | $\sigma = 0.25$ | 72.51 | 59.25 | 43.64 | 28.25 | 0.0 | 0.0 | 0.0 | 0.0 | 0.0 | 0.0 | 0.0 |
| | $\sigma = 0.50$ | 61.23 | 52.52 | 43.44 | 34.65 | 26.57 | 19.39 | 13.0 | 7.5 | 0.0 | 0.0 | 0.0 |
| **MACER-DS** | $\sigma = 0.12$  K=100 | 79.21 | 60.57 | 30.95 | 0.0 | 0.0 | 0.0 | 0.0 | 0.0 | 0.0 | 0.0 | 0.0 |
| | $\sigma = 0.12$  K=200 | 79.3 | 60.98 | 30.18 | 0.0 | 0.0 | 0.0 | 0.0 | 0.0 | 0.0 | 0.0 | 0.0 |
| | $\sigma = 0.12$  K=300 | 79.39 | 61.33 | 27.9 | 0.07 | 0.0 | 0.0 | 0.0 | 0.0 | 0.0 | 0.0 | 0.0 |
| | $\sigma = 0.12$  K=400 | 79.45 | 61.27 | 25.62 | 10.07 | 0.0 | 0.0 | 0.0 | 0.0 | 0.0 | 0.0 | 0.0 |
| | $\sigma = 0.12$  K=500 | 79.48 | 61.4 | 23.43 | 11.02 | 0.01 | 0.0 | 0.0 | 0.0 | 0.0 | 0.0 | 0.0 |
| | $\sigma = 0.12$  K=600 | 79.44 | 61.55 | 22.22 | 10.66 | 0.11 | 0.0 | 0.0 | 0.0 | 0.0 | 0.0 | 0.0 |
| | $\sigma = 0.12$  K=700 | 79.5 | 61.39 | 21.79 | 9.94 | 3.82 | 0.0 | 0.0 | 0.0 | 0.0 | 0.0 | 0.0 |
| | $\sigma = 0.12$  K=800 | 79.47 | 61.25 | 21.83 | 9.33 | 5.38 | 0.0 | 0.0 | 0.0 | 0.0 | 0.0 | 0.0 |
| | $\sigma = 0.12$  K=900 | 79.48 | 61.34 | 21.59 | 8.89 | 6.02 | 0.1 | 0.0 | 0.0 | 0.0 | 0.0 | 0.0 |
| | $\sigma = 0.25$  K=100 | 73.41 | 63.59 | 46.37 | 27.96 | 12.76 | 0.0 | 0.0 | 0.0 | 0.0 | 0.0 | 0.0 |
| | $\sigma = 0.25$  K=200 | 73.72 | 65.1 | 47.51 | 27.19 | 13.85 | 0.0 | 0.0 | 0.0 | 0.0 | 0.0 | 0.0 |
| | $\sigma = 0.25$  K=300 | 73.9 | 65.63 | 47.81 | 26.42 | 13.19 | 0.0 | 0.0 | 0.0 | 0.0 | 0.0 | 0.0 |
| | $\sigma = 0.25$  K=400 | 73.96 | 66.03 | 48.12 | 25.14 | 12.2 | 4.17 | 0.0 | 0.0 | 0.0 | 0.0 | 0.0 |
| | $\sigma = 0.25$  K=500 | 74.0 | 66.18 | 47.97 | 23.98 | 11.01 | 4.59 | 0.0 | 0.0 | 0.0 | 0.0 | 0.0 |
| | $\sigma = 0.25$  K=600 | 74.04 | 66.41 | 48.23 | 23.4 | 9.74 | 4.23 | 0.0 | 0.0 | 0.0 | 0.0 | 0.0 |
| | $\sigma = 0.25$  K=700 | 74.02 | 66.47 | 48.18 | 22.86 | 8.65 | 3.78 | 0.0 | 0.0 | 0.0 | 0.0 | 0.0 |
| | $\sigma = 0.25$  K=800 | 74.07 | 66.68 | 48.12 | 22.58 | 7.62 | 3.25 | 1.06 | 0.0 | 0.0 | 0.0 | 0.0 |
| | $\sigma = 0.25$  K=900 | 74.01 | 66.74 | 48.24 | 22.37 | 6.88 | 2.74 | 1.08 | 0.0 | 0.0 | 0.0 | 0.0 |
| | $\sigma = 0.50$  K=100 | 62.62 | 55.99 | 47.65 | 38.37 | 28.3 | 19.54 | 12.75 | 7.55 | 0.0 | 0.0 | 0.0 |
| | $\sigma = 0.50$  K=200 | 63.07 | 57.27 | 49.54 | 40.25 | 29.36 | 19.44 | 12.35 | 7.43 | 3.23 | 0.0 | 0.0 |
| | $\sigma = 0.50$  K=300 | 63.28 | 57.91 | 50.46 | 41.4 | 30.0 | 19.41 | 11.99 | 7.08 | 3.54 | 0.0 | 0.0 |
| | $\sigma = 0.50$  K=400 | 63.39 | 58.25 | 51.18 | 41.98 | 30.22 | 19.11 | 11.69 | 6.9 | 3.97 | 0.0 | 0.0 |
| | $\sigma = 0.50$  K=500 | 63.5 | 58.51 | 51.51 | 42.4 | 30.66 | 18.7 | 11.13 | 6.73 | 3.83 | 1.06 | 0.0 |
| | $\sigma = 0.50$  K=600 | 63.57 | 58.72 | 51.83 | 42.62 | 30.51 | 18.66 | 10.85 | 6.44 | 3.7 | 1.61 | 0.0 |
| | $\sigma = 0.50$  K=700 | 63.65 | 58.9 | 52.06 | 42.79 | 30.63 | 18.25 | 10.57 | 6.25 | 3.53 | 1.67 | 0.0 |
| | $\sigma = 0.50$  K=800 | 63.74 | 59.02 | 52.19 | 42.96 | 30.62 | 18.2 | 10.18 | 5.84 | 3.35 | 1.66 | 0.45 |
| | $\sigma = 0.50$  K=900 | 63.79 | 59.09 | 52.28 | 43.03 | 30.75 | 18.21 | 9.89 | 5.53 | 3.13 | 1.62 | 0.52 |
| **MACER-DS$^2$ (n=1)** | $\sigma = 0.12$  K=100 | 79.57 | 61.25 | 34.66 | 0.0 | 0.0 | 0.0 | 0.0 | 0.0 | 0.0 | 0.0 | 0.0 |
| | $\sigma = 0.12$  K=200 | 79.58 | 61.57 | 36.29 | 0.0 | 0.0 | 0.0 | 0.0 | 0.0 | 0.0 | 0.0 | 0.0 |
| | $\sigma = 0.12$  K=300 | 79.42 | 61.35 | 36.21 | 0.06 | 0.0 | 0.0 | 0.0 | 0.0 | 0.0 | 0.0 | 0.0 |
| | $\sigma = 0.12$  K=400 | 79.44 | 61.1 | 35.32 | 12.32 | 0.0 | 0.0 | 0.0 | 0.0 | 0.0 | 0.0 | 0.0 |
| | $\sigma = 0.12$  K=500 | 79.2 | 60.64 | 34.22 | 13.65 | 0.0 | 0.0 | 0.0 | 0.0 | 0.0 | 0.0 | 0.0 |
| | $\sigma = 0.12$  K=600 | 79.09 | 60.23 | 33.75 | 13.25 | 0.0 | 0.0 | 0.0 | 0.0 | 0.0 | 0.0 | 0.0 |
| | $\sigma = 0.12$  K=700 | 78.98 | 60.01 | 32.89 | 12.66 | 1.46 | 0.0 | 0.0 | 0.0 | 0.0 | 0.0 | 0.0 |
| | $\sigma = 0.12$  K=800 | 78.85 | 59.65 | 32.65 | 12.07 | 2.24 | 0.0 | 0.0 | 0.0 | 0.0 | 0.0 | 0.0 |
| | $\sigma = 0.12$  K=900 | 78.78 | 59.52 | 32.3 | 11.4 | 2.25 | 0.0 | 0.0 | 0.0 | 0.0 | 0.0 | 0.0 |
| | $\sigma = 0.12$  K=1000 | 78.73 | 59.15 | 31.58 | 10.63 | 2.05 | 0.0 | 0.0 | 0.0 | 0.0 | 0.0 | 0.0 |
| | $\sigma = 0.25$  K=100 | 71.45 | 59.44 | 45.71 | 30.76 | 14.57 | 0.0 | 0.0 | 0.0 | 0.0 | 0.0 | 0.0 |
| | $\sigma = 0.25$  K=200 | 71.81 | 60.13 | 46.5 | 31.3 | 16.2 | 0.0 | 0.0 | 0.0 | 0.0 | 0.0 | 0.0 |
| | $\sigma = 0.25$  K=300 | 71.81 | 60.13 | 46.5 | 31.3 | 16.2 | 0.0 | 0.0 | 0.0 | 0.0 | 0.0 | 0.0 |
| | $\sigma = 0.25$  K=400 | 71.91 | 60.48 | 46.51 | 30.83 | 16.73 | 5.66 | 0.0 | 0.0 | 0.0 | 0.0 | 0.0 |
| | $\sigma = 0.25$  K=500 | 71.84 | 60.56 | 46.3 | 30.26 | 16.43 | 7.07 | 0.0 | 0.0 | 0.0 | 0.0 | 0.0 |
| | $\sigma = 0.25$  K=600 | 71.77 | 60.38 | 45.92 | 29.84 | 16.11 | 7.14 | 0.0 | 0.0 | 0.0 | 0.0 | 0.0 |
| | $\sigma = 0.25$  K=700 | 71.69 | 60.12 | 45.66 | 29.41 | 15.6 | 7.0 | 0.04 | 0.0 | 0.0 | 0.0 | 0.0 |
| | $\sigma = 0.25$  K=800 | 71.73 | 60.19 | 45.41 | 28.91 | 15.07 | 6.68 | 2.15 | 0.0 | 0.0 | 0.0 | 0.0 |
| | $\sigma = 0.25$  K=900 | 71.68 | 60.11 | 45.14 | 28.51 | 14.54 | 6.23 | 2.34 | 0.0 | 0.0 | 0.0 | 0.0 |
| | $\sigma = 0.25$  K=1000 | 71.63 | 59.98 | 44.97 | 28.21 | 14.08 | 5.9 | 2.12 | 0.0 | 0.0 | 0.0 | 0.0 |
| | $\sigma = 0.50$  K=100 | 60.96 | 53.69 | 44.96 | 36.64 | 28.13 | 20.46 | 14.44 | 8.73 | 0.01 | 0.0 | 0.0 |
| | $\sigma = 0.50$  K=200 | 61.37 | 54.35 | 46.07 | 37.43 | 28.55 | 20.58 | 14.26 | 8.65 | 3.6 | 0.0 | 0.0 |
| | $\sigma = 0.50$  K=300 | 61.52 | 54.74 | 46.53 | 37.9 | 28.91 | 20.62 | 14.12 | 8.42 | 3.9 | 0.0 | 0.0 |
| | $\sigma = 0.50$  K=400 | 61.42 | 54.81 | 46.83 | 38.02 | 28.98 | 20.51 | 13.69 | 8.3 | 4.27 | 0.0 | 0.0 |
| | $\sigma = 0.50$  K=500 | 61.39 | 54.74 | 47.03 | 38.2 | 28.85 | 20.25 | 13.45 | 8.14 | 4.16 | 1.0 | 0.0 |
| | $\sigma = 0.50$  K=600 | 61.44 | 54.8 | 46.96 | 38.2 | 28.83 | 19.97 | 13.23 | 7.94 | 4.1 | 1.53 | 0.0 |
| | $\sigma = 0.50$  K=700 | 61.35 | 54.75 | 46.89 | 38.04 | 28.7 | 19.53 | 12.95 | 7.64 | 3.98 | 1.7 | 0.0 |
| | $\sigma = 0.50$  K=800 | 61.24 | 54.75 | 46.94 | 38.1 | 28.49 | 19.3 | 12.59 | 7.46 | 3.9 | 1.69 | 0.4 |
| | $\sigma = 0.50$  K=900 | 61.25 | 54.73 | 46.85 | 37.94 | 28.19 | 18.87 | 12.29 | 7.13 | 3.69 | 1.7 | 0.51 |
| | $\sigma = 0.50$  K=1000 | 61.21 | 54.72 | 46.84 | 37.87 | 27.97 | 18.74 | 12.08 | 6.82 | 3.43 | 1.66 | 0.71 |

Table 10: **Certified accuracy per radius on CIFAR10.** We report MACER-DS$^2(n = 8)$ under varying $\sigma$ and number of iterations $K$ in Algorithm 1.

| | $\ell_2^r$ (CIFAR10) | | 0.0 | 0.25 | 0.50 | 0.75 | 1.00 | 1.25 | 1.50 | 1.75 | 2.00 | 2.25 | 2.50 |
|---|---|---|---|---|---|---|---|---|---|---|---|---|---|
| | $\sigma$ =0.12 | K=100 | 81.9 | 62.52 | 29.38 | 0.0 | 0.0 | 0.0 | 0.0 | 0.0 | 0.0 | 0.0 | 0.0 |
| | $\sigma$ =0.12 | K=200 | 82.16 | 63.09 | 29.72 | 0.0 | 0.0 | 0.0 | 0.0 | 0.0 | 0.0 | 0.0 | 0.0 |
| | $\sigma$ =0.12 | K=300 | 82.2 | 63.4 | 28.47 | 0.0 | 0.0 | 0.0 | 0.0 | 0.0 | 0.0 | 0.0 | 0.0 |
| | $\sigma$ =0.12 | K=400 | 82.21 | 63.51 | 26.29 | 6.84 | 0.0 | 0.0 | 0.0 | 0.0 | 0.0 | 0.0 | 0.0 |
| | $\sigma$ =0.12 | K=500 | 82.34 | 63.76 | 24.13 | 7.62 | 0.0 | 0.0 | 0.0 | 0.0 | 0.0 | 0.0 | 0.0 |
| | $\sigma$ =0.12 | K=600 | 82.34 | 63.66 | 22.6 | 7.05 | 0.0 | 0.0 | 0.0 | 0.0 | 0.0 | 0.0 | 0.0 |
| | $\sigma$ =0.12 | K=700 | 82.32 | 63.84 | 21.61 | 6.48 | 0.6 | 0.0 | 0.0 | 0.0 | 0.0 | 0.0 | 0.0 |
| | $\sigma$ =0.12 | K=800 | 82.35 | 63.9 | 21.06 | 5.4 | 0.83 | 0.0 | 0.0 | 0.0 | 0.0 | 0.0 | 0.0 |
| | $\sigma$ =0.12 | K=900 | 82.37 | 63.9 | 20.79 | 4.67 | 0.82 | 0.0 | 0.0 | 0.0 | 0.0 | 0.0 | 0.0 |
| | $\sigma$ =0.12 | K=1000 | 82.39 | 63.89 | 20.57 | 3.88 | 0.77 | 0.0 | 0.0 | 0.0 | 0.0 | 0.0 | 0.0 |
| MACER-DS$^2$ (n=8) | $\sigma$ =0.25 | K=100 | 75.18 | 64.79 | 47.14 | 29.31 | 12.56 | 0.0 | 0.0 | 0.0 | 0.0 | 0.0 | 0.0 |
| | $\sigma$ =0.25 | K=200 | 75.36 | 66.23 | 48.55 | 28.91 | 13.99 | 0.0 | 0.0 | 0.0 | 0.0 | 0.0 | 0.0 |
| | $\sigma$ =0.25 | K=300 | 75.53 | 66.87 | 49.24 | 28.31 | 14.26 | 0.01 | 0.0 | 0.0 | 0.0 | 0.0 | 0.0 |
| | $\sigma$ =0.25 | K=400 | 75.57 | 67.36 | 49.53 | 27.35 | 13.94 | 3.86 | 0.0 | 0.0 | 0.0 | 0.0 | 0.0 |
| | $\sigma$ =0.25 | K=500 | 75.65 | 67.71 | 49.59 | 26.24 | 13.23 | 4.68 | 0.01 | 0.0 | 0.0 | 0.0 | 0.0 |
| | $\sigma$ =0.25 | K=600 | 75.72 | 67.81 | 49.64 | 25.46 | 12.12 | 4.73 | 0.01 | 0.0 | 0.0 | 0.0 | 0.0 |
| | $\sigma$ =0.25 | K=700 | 75.84 | 67.93 | 49.85 | 24.92 | 11.1 | 4.58 | 0.01 | 0.01 | 0.0 | 0.0 | 0.0 |
| | $\sigma$ =0.25 | K=800 | 75.81 | 68.08 | 49.84 | 24.79 | 10.18 | 4.33 | 1.05 | 0.01 | 0.0 | 0.0 | 0.0 |
| | $\sigma$ =0.25 | K=900 | 75.87 | 68.16 | 49.84 | 24.53 | 9.38 | 3.81 | 1.02 | 0.01 | 0.0 | 0.0 | 0.0 |
| | $\sigma$ =0.25 | K=1000 | 75.87 | 68.26 | 49.94 | 24.27 | 8.66 | 3.32 | 0.93 | 0.01 | 0.01 | 0.0 | 0.0 |
| | $\sigma$ =0.50 | K=100 | 61.79 | 55.41 | 47.8 | 39.03 | 29.04 | 20.62 | 13.83 | 7.92 | 0.0 | 0.0 | 0.0 |
| | $\sigma$ =0.50 | K=200 | 62.11 | 56.53 | 49.36 | 40.68 | 30.08 | 20.55 | 13.38 | 7.84 | 3.07 | 0.0 | 0.0 |
| | $\sigma$ =0.50 | K=300 | 62.31 | 57.21 | 50.54 | 41.6 | 30.79 | 20.46 | 13.03 | 7.56 | 3.44 | 0.0 | 0.0 |
| | $\sigma$ =0.50 | K=400 | 62.49 | 57.68 | 51.12 | 42.08 | 31.12 | 20.21 | 12.45 | 7.34 | 3.65 | 0.0 | 0.0 |
| | $\sigma$ =0.50 | K=500 | 62.62 | 57.98 | 51.61 | 42.41 | 31.3 | 20.13 | 12.09 | 6.94 | 3.65 | 0.74 | 0.0 |
| | $\sigma$ =0.50 | K=600 | 62.71 | 58.28 | 51.82 | 42.85 | 31.52 | 19.78 | 11.58 | 6.54 | 3.56 | 1.28 | 0.0 |
| | $\sigma$ =0.50 | K=700 | 62.84 | 58.35 | 52.15 | 43.04 | 31.42 | 19.6 | 11.12 | 6.33 | 3.29 | 1.37 | 0.0 |
| | $\sigma$ =0.50 | K=800 | 62.91 | 58.45 | 52.33 | 43.29 | 31.48 | 19.47 | 10.62 | 5.95 | 3.21 | 1.39 | 0.27 |
| | $\sigma$ =0.50 | K=900 | 62.93 | 58.54 | 52.56 | 43.44 | 31.49 | 19.14 | 10.17 | 5.73 | 3.09 | 1.34 | 0.38 |
| | $\sigma$ =0.50 | K=1000 | 63.0 | 58.66 | 52.67 | 43.47 | 31.66 | 19.04 | 9.85 | 5.49 | 2.85 | 1.21 | 0.4 |

Table 11: **Best certified accuracy per $\ell_1$ radii and ACR** of YANG and YANG-DS.

| $\ell_1^r$(CIFAR10) | 0.0 | 0.25 | 0.5 | 0.75 | 1.0 | 1.5 | 2.0 | ACR |
|---|---|---|---|---|---|---|---|---|
| YANG | 92 | 83 | 75 | 71 | 46 | 0 | 0 | 0.775 |
| YANG-DS | 92 | **89** | **82** | **76** | **58** | **6** | **2** | **0.946** |

| $\ell_1^r$(ImageNet) | 0.0 | 0.25 | 0.5 | 0.75 | 1.0 | 1.5 | 2.0 | ACR |
|---|---|---|---|---|---|---|---|---|
| YANG | 78 | 73 | 67 | 63 | 0 | 0 | 0 | 0.683 |
| YANG-DS | **79** | **76** | **70** | **65** | **46** | 0 | 0 | **0.729** |

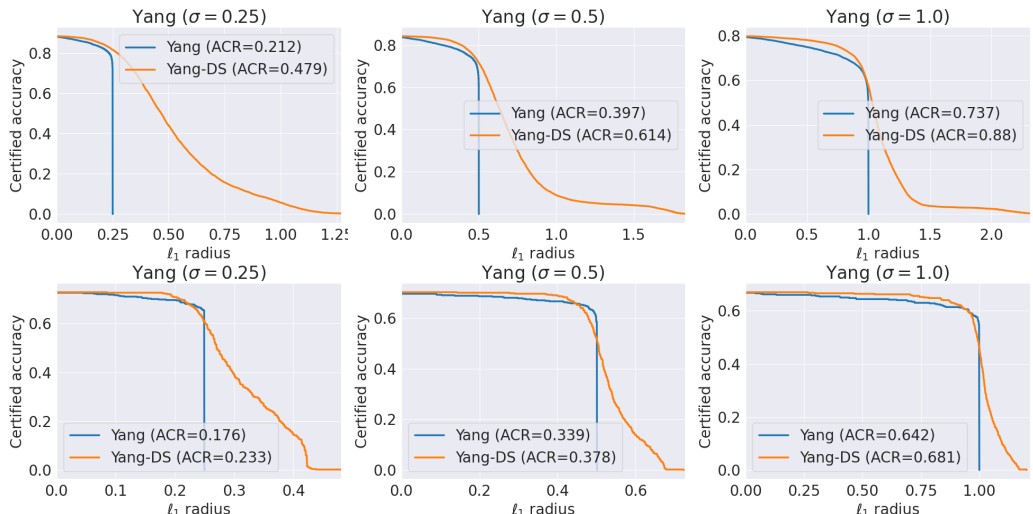

Figure 11: $\ell_1$ **Certified accuracy comparison against Yang per radius per $\sigma$.** We compare Yang against Yang-DS. We show CIFAR10 and ImageNet results in first and second rows, respectively. Similar to the earlier experiments on $\ell_2$ certificate, deploying data-dependent smoothing with the memory enhanced classifier yields significant improvement for the $\ell_1$ certified accuracy in all considered scenarios.

# G  DATA-DEPENDENT SMOOTHING FOR $\ell_1$ CERTIFICATES.

While indeed we focused both our methodology and experiments on $\ell_2$ certificates, our methodology is extendable to any other $\ell_p$ certificate. For that regard and as per reviewer's request, we conducted experiments on $\ell_1$ certification. We leveraged the results of Yang et al. (2020) that derived the tightest $\ell_1$ certificate using randomized smoothing with uniform distribution $\mathcal{U}[-\lambda, \lambda]^d$. The certified radius in that case has the form $\mathcal{R}_1 = \lambda(p_A - p_B)$. We replace our objective in Equation equation 3 with:

$$\lambda_x^* = \arg\max_\lambda \lambda \left( \mathbb{E}_{\epsilon \sim \mathcal{U}[-\lambda,\lambda]^d}(f_\theta^{c_A}(x+\epsilon)) - \max_{c \neq c_A} \mathbb{E}_{\epsilon \sim \mathcal{U}[-\lambda,\lambda]^d}(f_\theta^c(x+\epsilon)) \right).$$

We solved our objective in an identical fashion to our Algorithm 1 with the same hyperparameters for $\lambda \in \{0.25, 0.5, 1.0\}$ in certification on both CIFAR10 and ImageNet. Further, we combine our data-dependent smooth classifier with the memory based algorithm proposed in Section 3.5. It is worthwhile mentioning that similar to the $\ell_2$ case, the memory based algorithm did not find any overlap between the certified regions of any pair of instances. We report the results in Figure 11 and Table 11. We observe that, similar to our extensive experiments on the $\ell_2$ certificate, our proposed memory-enhanced data-dependent smoothing yields consistent improvement in the $\ell_1$ certified accuracy. We report an improvement of 7% and 3% over the state of the art certified accuracy at $\ell_1$ radius of 0.5 on CIFAR10 and ImageNet, respectively. At last, we note similar improvement to the $\ell_1$ ACR as reported in Table 11.

## H  Runtime

We measure the certification runtime on an NVIDIA Quadro RTX-6000 GPU for our proposed data dependent smoothed classifier (time includes Algorithm 1 in addition to the memory based certification) compared to the certification of a fixed $\sigma$ classifier. Certifying one CIFAR10 test input with ResNet18 takes 1.6 and an average of 1.8 seconds for a fixed $\sigma$ classifier and for the data dependent classifier ($K = 900$), respectively. Certifying an ImageNet test input on ResNet50 takes 109.5 and an average of 136 seconds for a fixed $\sigma$ classifier and our data dependent classifier ($K = 400$), respectively. The runtime overhead added by using Algorithm 1 and memory based certification is negligible compared to the gains in certified accuracy.

