# OpenReview forum: "Data-Dependent Randomized Smoothing"
_ICLR.cc/2022/Conference — ICLR 2022 Submitted_

### Official Review · Reviewer_DGVf · 2021-10-27

**Correctness:** 4
**Technical Novelty And Significance:** 2
**Empirical Novelty And Significance:** 2
**Recommendation:** 6
**Confidence:** 3

**Main Review:**

The paper is well written and the experiments are well developed.

The main contribution of the paper is to adapt the smoothing parameter $\sigma$ to every input data point by maximizing the certification radius $R$. This is a natural way to improve the randomized smoothing technique. Some questions are given below.

1. If one data $x$ is close to the boundary and classified incorrectly (e.g., suppose $x_3$ in Fig.1 is classified incorrectly), then in this case optimizing the radius would actually reduce the value of $\sigma$. So will this approach apply to models that have low classification accuracy?

2. Regarding the runtime/scalability. If we apply this data-dependent randomized smoothing to perform adversarial training, it seems that we need to generate a specialized gaussian noise for every single sample at each time. In the SmoothAdv algorithm, there they can reuse the gaussian noises to corrupt a batch of samples (because $\sigma$ is fixed). Can the authors comment on the scalability of your approach?

3. The paper discusses $\ell_2^r$ certification, can the proposed approach extends to more general types of certifications?

**Summary Of The Paper:**

This paper proposes a new randomized smoothing technique to improve the robustness of models. Specifically, in the original randomized smoothing approach, we use a fixed variance parameter $\sigma$ for the Gaussian smoothing. The authors claim that this is in general suboptimal as some data samples far away from the decision boundary can be further smoothed. Motivated by this, the authors propose to optimize the smoothing parameter $\sigma$ for every data sample, by solving an optimization problem using Monte Carlo approximation and SGD. The paper also provides a simple procedure for the certification of the data dependent smooth classifier. Through extensive experiments, the paper shows the effectiveness of the proposed approach.

**Summary Of The Review:**

See the comments.

---

> ### Author Response · Authors · 2021-11-13
> **Response**
>
> **Our response.** We thank the reviewer for the detailed feedback. We address next the questions raised by the reviewer.
>
> **Regarding the applicability of our approach to models with low accuracy.** Note that our approach for certification does not depend on the ground truth label but aims to maximize the certification radius of the top prediction. That is to say, our approach will increase the certified accuracy of the \emph{correctly} classified instances only. Thus, our approach can generally be applied to models with low classification accuracy but can only improve the certified accuracy by improving the radius of those few correctly classified instances. We would like to mention that this property is inherited from the formula of the certified radius since it depends on the top 2 predictions $c_A$ and $c_B$ while being agnostic from the ground truth label.
>
>
>
> **Regarding the runtime and scalability.** When deploying data-dependent smoothing in training and certification, we update the smoothing parameters with $n, K = 1$ in Algorithm 1 at each training epoch. This will require an additional forward and backward pass per training iteration resulting in doubling the total training computation. Note that for SmoothAdv which deploys adversarial training, we first optimize the smoothing parameter $\sigma$ for a given input. This determines the smooth classifier that shall be adversarially trained with the Gaussian samples. We sample the Gaussian noise from the optimized distribution once and use those same exact samples during the attack iterations throughout the training procedure. Thus, our additional computational requirement will be similar to the earlier setup: an additional forward an backward pass per training iteration.
>
>
>
> **Regarding the extension to other certificates.** While indeed we focused both our methodology and experiments on $\ell_2$ certificates, our methodology is extendable to any other $\ell_p$ certificate. For that regard and as per reviewer's request, we conducted experiments on $\ell_1$ certification. We leveraged the results of Yang et.al. [A] that derived the tightest $\ell_1$ certificate using randomized smoothing with uniform distribution $\mathcal U[-\lambda, \lambda]^d$. The certified radius in that case has the form $\mathcal R_1 = \lambda(p_A - p_B)$. We replace our objective in Equation (2) with:
> $$
> \lambda_x^* = \text{arg}\max_\lambda \lambda \left(\mathbb E_{\epsilon\sim \mathcal{U}[-\lambda, \lambda]^d}(f_\theta^{c_A}(x+\epsilon)) - \max_{c \neq c_A} \mathbb E_{\epsilon\sim \mathcal{U}[-\lambda, \lambda]^d}(f_\theta^{c}(x+\epsilon))  \right).
> $$
>
> We leave Algorithm 1 followed by the memory based certification (Algorithm 3) unchanged where we set the hyperparameters $\lambda \in \{0.25, 0.5, 1.0\}$ in certification identically for both CIFAR10 and ImageNet. We report the best certified accuracy over $\lambda$ used in certification at each radius in the table below in a similar fashion to Tables 1, 2 and 3. Moreover, we plot the certified accuracy for each considered $\lambda$ in the newly added Appendix G. Similarly to the certified accuracy results with $\ell_2$, our memory-enhanced data dependent classifier provides significant improvement in the $\ell_1$ certificate too over the state-of-the-art $\ell_1$ certificate of Yang e. al. [A]. We updated our paper to include these results in Appendix G.
>
>
> [A] Greg Yang, Tony Duan, Edward Hu, Hadi Salman, Ilya Razenshteyn, and Jerry Li.  Randomized smoothing of all shapes and sizes. International Conference on Machine Learning (ICML), 2020
>
>
>
> | $\ell_1^r$ (CIFAR10) | 0.0 | 0.25 | 0.5 | 0.75 | 1.0 | 1.5 | 2.0 |  ACR  |
> |:-------------------:|:---:|:----:|:-------:|:----:|:---:|:---:|:---:|:-----:|
> |     Yang et al.     |  92 |  83  |    75   |  71  |  46 |  0  |  0  | 0.775 |
> |   Yang et al. - DS  |  92 |  **89**  |    **82**   |  **76**  |  **58** |  **6**  | **2**  | **0.946** |
>
> | $\ell_1^r$ (ImageNet) | 0.0 | 0.25 | 0.5 | 0.75 | 1.0 | 1.5 | 2.0 |  ACR  |
> |:--------------------:|:---:|:----:|:-------:|:----:|:---:|:---:|:---:|:-----:|
> |      Yang et al.     |  78 |  73  |    67   |  63  |  0  |  0  |  0  | 0.683 |
> |   Yang et al. - DS   |  **79** |  **76**  |    **70**   |  **65**  |  **46** |  0  |  0  | **0.729** |

---

### Official Review · Reviewer_y9v6 · 2021-10-29

**Correctness:** 3
**Technical Novelty And Significance:** 3
**Empirical Novelty And Significance:** 3
**Recommendation:** 6
**Confidence:** 1

**Main Review:**

The paper is not in my area but I find it easy to read and follow. I agree that the making the variance of the Gaussian noise to be data-dependent has the potential to largely improve the performance of the randomized smoothing classifier. However, it is not clear to me that why maximizing the radius $R$ would be beneficial to the resulting smooth classifier. It would be better if the author(s) could provide more motivation and justification on their proposed objective function.

**Summary Of The Paper:**

The paper focuses on training robust deep neural networks. The author(s) developed a data-dependent randomized smoothing method to certify the DNN classifier. Their proposed methods are evaluated on benchmark datasets and are shown to outperform other baseline approaches.

**Summary Of The Review:**

The author(s) might want to discuss in detail why it is desirable to maximize the radius $R$ in their proposed objective function.

---

> ### Author Response · Authors · 2021-11-13
> **Response**
>
> **Our response.** We thank the reviewer for the feedback.
>
> **Regarding the aim behind maximizing the certified radius.** Our main motivation is to enhance the certified accuracy. To do so, one needs to improve model accuracy over all predicted samples or increase the region around every input under which the prediction is constant, i.e. maximizing R, or both jointly. Models with larger certified accuracy resist larger adversarial perturbations. This is desirable since as deep models are constantly been deployed in real world applications in which they need to be provably robust and reliable in practice against such additive adversaries.

---

### Official Review · Reviewer_Xsdx · 2021-10-31

**Correctness:** 3
**Technical Novelty And Significance:** 3
**Empirical Novelty And Significance:** 2
**Recommendation:** 5
**Confidence:** 4

**Main Review:**

The problem is important and the idea of this paper is interesting but seems quite simple, which simply optimizes the certifiable lower bound of randomized smoothing. I found it a big surprising that this has not been done before. Given the certifiable bound is already there, it seems there is not too much technical difficulties for this method, so I wonder why this has not been done before, given the randomized smoothing has been proposed for several years? Can the authors elaborate on this?

As far as I understand, the method actually seems to be only applicable to the l2 certification, given the bound is only for l2. So it seems a bit over claiming in the abstract that the method is generic. In addition to this, I believe there are bounds for the other norm certification, and I wonder if the proposed method can be applied to those cases? One example and missing reference is the paper by Li et al 2019: Certified Adversarial Robustness with Additive Noise. In that paper, they propose information theoretic analysis and get a more general bound (although the bound seems to be looser than the one by Cohen et al.), which can do L1 certification. I think a study for this would great improve the paper and should be performed empirically. In addition, Li et al also propose an enhanced training strategy called stability training, which is shown to be able to greatly improve the performance. I would suggest the authors to investigate that training strategy, in addition to the current ones using Cohen, smoothadv and macer.

Another comment is about the setting of the parameter n in Algorithm 1. It seems the authors only consider setting n = 1, which I think is highly inaccurate with high variance. Although the authors provide some results in the appendix with n=8, I think the study is not deep enough. More detailed studies on this parameter and the effect on the accuracy as well as the balance between training time and accurate need to be investigated.

A minor comment: the visualization in Figure 6 does not seem the provide much information. A better visualization I am looking for actually is similar to the illustration in Figure 1, but on a real case. I am expecting the decision boundary can be drawn and the optimized radiuses of different data samples can be shown as well.

**Summary Of The Paper:**

This papers proposes to optimize the certified bound of randomized smoothing to get data dependent perturbations for different data. It also proposes a simple memory-based approach to certifying the resultant smooth classifier due to the lack of guarantee on the overlapping of the perturbation regions. The technique is incorporated into 3 randomized smoothing approaches, which shows improvement over existing methods in experiments.

**Summary Of The Review:**

The idea of this paper is interesting and the paper itself is a solid work to improve model robustness by optimizing the certified bounds w.r.t. the noise variance. However, there are missing comparisons with existing techniques/methods, which I hope to be solved.

---

> ### Author Response · Authors · 2021-11-13
> **Response**
>
>
> **Our response.** We thank the reviewer for the detailed feedback. We address next the raised questions below.
>
> **Regarding the fact that this idea was not done before.** While several efforts investigated training better certified models through incorporating adversarial training [A] or explicitly maximizing the radius over parameters [B], we are not aware of any published work that improved the certified accuracy by optimizing the parameters of the smoothing distribution per input. We believe the reason why there has not been work in this direction is because the per input optimization of the smoothing distribution alone, i.e. without the memory consideration, does not yield a sound certification as discussed in Section 3.5. We alleviated this problem with the memory based Algorithm 2 proposed in Section 3.5 and detailed in Appendix C.
>
>
> **Regarding the applicability of our framework to other $\ell_p$ certificates.** While indeed we focused both our methodology and experiments on $\ell_2$ certificates, our methodology is extendable to any other $\ell_p$ certificate. For that regard and as per reviewer's request, we conducted experiments on $\ell_1$ certification. We leveraged the results of Yang et.al. [C] that derived the tightest $\ell_1$ certificate using randomized smoothing with uniform distribution $\mathcal U[-\lambda, \lambda]^d$. The certified radius in that case has the form $\mathcal R_1 = \lambda(p_A - p_B)$. We replace our objective in Equation (2) with:
>
> $$
> \lambda_x^* = \text{arg}\max_\lambda \lambda \left(\mathbb E_{\epsilon\sim \mathcal{U}[-\lambda, \lambda]^d}(f_\theta^{c_A}(x+\epsilon)) - \max_{c \neq c_A} \mathbb E_{\epsilon\sim \mathcal{U}[-\lambda, \lambda]^d}(f_\theta^{c}(x+\epsilon))  \right).
> $$
>
> We leave Algorithm 1 followed by the memory based certification (Algorithm 3) unchanged where we set the hyperparameters $\lambda \in$ {$0.25, 0.5, 1.0$} in certification identically for both CIFAR10 and ImageNet. We report the best certified accuracy over $\lambda$ used in certification at each radius in the table below in a similar fashion to Tables 1, 2 and 3. Moreover, we plot the certified accuracy for each considered $\lambda$ in the newly added Appendix G. Similarly to the certified accuracy results with $\ell_2$, our memory-based enhanced data-dependent classifier provides significant improvement in the $\ell_1$ certificate too over the state-of-the-art $\ell_1$ certificate of Yang e. al. [C]. We updated our paper to include these results in Appendix G. Moreover, we cited the missing reference of Li et. al. in our related work section.
>
>
> **Regarding the choice of $n=1$.** While the choice of $n=1$ might lead to high variance, we alleviated this challenge by setting the learning rate, $\alpha$ in Algorithm 1, to be small. The main motive behind setting $n=1$ is that higher values of $n$ will impose higher computational requirements where we only need a crude estimate for $\sigma_x$ for an improved certification, i.e. we do not aim to solve the exact optimal $\sigma_x$. Nevertheless, we reported results with $n=8$ in the appendix where we observed very marginal effect on the certified accuracy. Regarding the effect on runtime, we have included the effect of $n$ on the added complexity in Appendix E.
>
>
>
> **Regarding the visualization of Figure 6.** Due to the high dimensionality of the datasets, we are unable to visualize a similar scenario to the one showed in Figure 1. We are also not sure how to select points that are close to the projected decision boundaries in 2D. We further note that the main aim of Figures 6 and 10 is to show the large variation in the optimal $\sigma^*_x$ as compared to setting one arbitrary.
>
>
>
>
> [A] "Provably robust deep learning via adversarially trained smoothed classifiers". Hadi Salman, Jerry Li, Ilya Razenshteyn, Pengchuan Zhang, Huan Zhang, Sebastien Bubeck, and
> Greg Yang. NeurIPS19.
>
> [B] "Macer: Attack-free and scalable robust training via maximizing certified radius". Runtian Zhai, Chen Dan, Di He, Huan Zhang, Boqing Gong, Pradeep Ravikumar, Cho-Jui Hsieh,
> and Liwei Wang. ICLR20.
>
> [C] "Randomized smoothing of all shapes and sizes". Greg Yang, Tony Duan, Edward Hu, Hadi Salman, Ilya Razenshteyn, and Jerry Li. ICML20.
>
>
>
> | $\ell_1^r$ (CIFAR10) | 0.0 | 0.25 | 0.5 | 0.75 | 1.0 | 1.5 | 2.0 |  ACR  |
> |:-------------------:|:---:|:----:|:-------:|:----:|:---:|:---:|:---:|:-----:|
> |     Yang et al.     |  92 |  83  |    75   |  71  |  46 |  0  |  0  | 0.775 |
> |   Yang et al. - DS  |  92 |  **89**  |    **82**   |  **76**  |  **58** |  **6**  | **2**  | **0.946** |
>
> | $\ell_1^r$ (ImageNet) | 0.0 | 0.25 | 0.5 | 0.75 | 1.0 | 1.5 | 2.0 |  ACR  |
> |:--------------------:|:---:|:----:|:-------:|:----:|:---:|:---:|:---:|:-----:|
> |      Yang et al.     |  78 |  73  |    67   |  63  |  0  |  0  |  0  | 0.683 |
> |   Yang et al. - DS   |  **79** |  **76**  |    **70**   |  **65**  |  **46** |  0  |  0  | **0.729** |

---

> > ### Comment · Reviewer_Xsdx · 2021-11-29
> > **boaderline**
> >
> > Thanks for the rebuttal. I am generally satisfied with most of the rebuttal. One remaining concern is on the impact of n to the algorithm. I still think n can impact the algorithm to some extent, while there is not a clear conclusion from the current results, i.e., it is hard to compare the results in the current form of multiple tables.
> >
> > Another issue after reading reviews from other reviewers is on the certificate of the proposed algorithm, as pointed out by Reviewer viFi: "The memory-based certification is changing the predictions of the (smooth) classifier. Thus, the final certificate is actually NOT a certificate for the original (smooth) classifier g -- but only for some adaptively changing classifier. " I agree with this, and the rebuttal says it is a certificate method based on the empirical results, which I think is not quite strictly correct. I would like to hear from Reviewer viFi on this. That being said, I think this is a borderline paper, and I am OK with it either being accepted or rejected.

---

> > > ### Author Response · Authors · 2021-11-29
> > > **Thank you**
> > >
> > > We thank the reviewer for the feedback.
> > >
> > > Regarding the effect of n, in the ablations we have conducted in the appendix, it does not seem to provide much gains with an immense overhead requirement for computational resources.
> > >
> > > Regarding the certificate that our method provides, this is true. Our certificate is provided for the memory enhanced data dependent smooth classifier. We have clarified this in text in the revised version based on the suggestions of viFi.
> > > Nevertheless, we hope that the reviewer does not find the clarification on the certification in text, which is resolved now, a ground reason for rejection. This is as all our results as demonstrated are sound with improved certified accuracy over several baselines as also acknowledged by Xsdx.

---

### Official Review · Reviewer_viFi · 2021-11-03

**Correctness:** 2
**Technical Novelty And Significance:** 3
**Empirical Novelty And Significance:** 3
**Recommendation:** 3
**Confidence:** 4

**Main Review:**

The paper tackles an important problem and aims to address one major limitation of the randomized smoothing framework: the constant variance of the (Gaussian) smoothing distribution. However, I am some concerns about the proposed approach:

1) The core idea of the approach is to optimize the sigma according to Eq. (2). However, this optimization principle assumes the constant-sigma set-up. Thus, I don't see why optimizing this equation is really principled.

Indeed, as described in Sec. 3.5, the resulting radius does NOT guarantee certification. (Side note: I find it a bit surprising to call the solution then the "radius of certification" even though it does not lead to a valid certification radius.)

The memory-based certification looks like an ad-hoc fix to this problem. This, however, leads to multiple further problems as described in the following.

2) The memory-based certification is changing the predictions of the (smooth) classifier. Thus, the final certificate is actually NOT a certificate for the original (smooth) classifier g -- but only for some adaptively changing classifier. Therefore, the original problem setting is actually never solved by the approach.

Moreover: Why should Eq. (2) make any sense for this classifier? Based on my understanding the memory-based principle could theoretically be used with any real number used as a radius. Thus, one could just randomly assign "radii of certification" to a new input sample. Put differently: The memory-based certificate is not really making any use of worst-case classifiers or alike.

Finally, since the certificate is given for this "changing classifier", I highly question the (practical) use of the certificate at all. The runtime complexity of the classifier -- and not just the certificate -- scales linearly with the number of samples seen. It is even worse than a KNN-classifier since one has to consider all test samples (and not only the training samples). Moreover, the outcome of the resulting (smooth) classifier will depend on the *order* how the test samples arrive. That is, the predictions might be different when a different order is used. Such behavior is very undesired for a classifier operating on iid data.

3) Given that the method is actually certifying some other classifier, I am also wondering whether the experimental comparison provides any insights. At the end, different classifiers are considered.


**Summary Of The Paper:**

The paper's core contribution is to extend randomized smoothing to a setting where the smoothing depends on the input data. That is, the variance of the smoothing distribution is not constant anymore but depends on the input. To ensure a proper certificate, the authors propose some memory-based technique (which essentially changes the final classifier based on the observed input data so far).

**Summary Of The Review:**

Overall, I highly question the use of the certificate. It is not certifying the actual smooth classifier as desired, but it certifies some other classifier (with highly undesired properties) in a non-scalable way. Thus, also the experimental comparison is questionable.

I am currently giving a score of 3. I might be open to increase it in case the authors clearly state that they are actually not certifying the smooth classifier. Put differently: The presentation has to be updated in various places since at the moment it is highly misleading and readers might get a wrong understanding of what is actually certified. Moreover, the authors should elaborate whether the experimental analysis is still useful or not.

---

> ### Author Response · Authors · 2021-11-13
> **Response**
>
> **Our response.** We thank the reviewer for the detailed feedback. Next, we address the raised concerns from the reviewer in the same order.
>
> **Regarding why the optimization in Eq. (2) is principled.** Our optimization in Eq. (2) is performed per input $x$ to find the optimal smoothing parameters for a given base classifier $f$. The solution is indeed a radius of certification for the input $x$ if $\sigma_x^*$ is used globally. While the data-dependent smooth classifier does not guarantee that, the equipped version with the memory based algorithm fixes the smoothing parameters for every certified region. This makes the optimized quantity, or an adjusted version of it a certified radius. We adjusted the text in Sections 1 and 3.5 to make it clearer.
>
> **Regarding certifying the memory-enhanced data dependent smooth classifier.** Indeed, the certificates are only on the the memory-enhanced data dependent smooth classifier and not on the data-dependent smooth classifier. However and as mentioned in Section 3.5, and for all datasets we conducted experiments on, our memory-based algorithm never found any intersection in the certified regions between differently predicted inputs (never fallen in the middle and right scenarios of Figure 2). That is to say, the reported certified accuracies for the memory-enhanced data dependent smooth classifier and the data-dependent smooth classifier are identical.
>
> **Regarding why should Eq. (2) make sense for our memory enhanced smooth classifier.** While it is the case that the memory certification is sound-by-construction for any radius, the choice of the radius of certification for every input is key for attaining a high certified accuracy. For example, if one were to take small radii around each point, the accuracy at such small radii might be close to the clean accuracy, but the overall classifier will only enjoy a small certified accuracy. On the other hand, arbitrarily selecting large radii for all inputs, this could lead initially to robust classifiers around a handful of points that are accurate. However, as more points are to be certified in memory, the accuracy will afterwards be severely affected dropping the overall certified accuracy. This is since many points will have their predictions incorrectly altered as they fall inside the prediction of other regions. Moreover, the overall certified accuracy will be affected as many other new points to be certified will have their certification radius reduced due to many intersections with previous memory samples. As such, a balance around each point is required. In our method, we propose to obtain such a balance through the use of *Certify* on our smooth classifier with $\sigma_x^*$ which leads to higher accuracy at each radii than previous methods.
>
>
> **Regarding the running time for our memory-enhanced classifier.** The reviewer is correct about the running time. We had a discussion in Appendix E as part of our main limitation, i.e. the running time scales linearly. However and as discussed in the runtime paragraph in Section 4.4, for all of our reported experiments on the datasets, the additional complexity inferred by our memory-based algorithm was rather marginal compared to the total complexity of the certification using randomized smoothing.
>
> **Regarding the dependence on order and the validity of our experimental comparisons.** We thank the reviewer for this keen observation. While indeed our memory-based classifier depends on the order at which the data is presented at test time, this is not the case if there is no overlap between the certified regions of differently predicted inputs. That is to say, if the middle and right scenarios of Figure 2 do not occur for any input, the memory based certification is order invariant. Note that as mentioned in Section 3.5, we found that this is the case in all of our experimental results where no instances with differently predicted labels share any overlap in their certified regions making our memory-enhanced classifier order invariant. That is, our results are directly comparable with all reported baselines. We added a discussion on this in Appendix E.
>
> At last and as per the recommendation of the reviewer, we adjusted the text in our paper (section 1 and 3 in the main paper and Appendix E) to better highlight that we certify the *memory-enhanced data dependent smooth classifier*. All newly added texts are in red. Regarding the experimental comparison: as per our response, all our reported results are order invariant making them directly comparable to all reported methods. We also added experiments for certification on $\ell_1$ norm in the appendix.

---

> > ### Comment · Reviewer_viFi · 2021-11-29
> > **Feedback**
> >
> > Thanks for your reply.
> >
> > Let me elaborate why Eq. (2) still bothers me: As you correctly state in your reply, it "holds" when using the same sigma globally. However, this is exactly what you are NOT doing. Thus, for me, Eq. (2) is to some degree arbitrary and one could also use some other function to optimize over. You could, e.g., take the maximizer of Eq. (2) and multiple it by 2. (This would not be completely random, will increase the radius of certification a lot, and will likely also not lead to overlapping regions.) I hope with this simple example, my point becomes clear?
> >
> > I think we agree that you are certifying the memory-based classifier. I am not happy how this is written in the revised version, though. For example, on page 1 and 2, you are still introducing the function g(x) -- giving the impression that this is the function/classifier you are certifying. However, this is not the case. Indeed, you have to define some g(x) which takes your memory into account.
> > (Even if for your experimental analysis, you have not observed any overlap, you are not certifying g(x))
> >
> > Running time and dependence on order: While in your experiments you might not have observed any problems, you cannot prevent this in real applications where you have to process millions of predictions. Also the runtime will surely explode then. It might be very subjective, but I am not convinced that this is a classifier someone wants to use.

---

### Author Response · Authors · 2021-11-15
**General Comment**

We would like to thank all reviewers for their time and efforts spent reviewing our paper. To address the reviewers concerns, we have made the following set of updates to the paper that are all marked in red:

- We emphasized in abstract, introduction, and section 3.5 that we certify a memory-enhanced data dependent smooth classifier as opposed to certifying the data dependent smooth classifier. Moreover, we added a discussion on why both the memory enhanced data dependent smooth classifier and the data dependent smooth classifier have identical performance under our setup.

- In Appendix C, we added a discussion on why our proposed data dependent optimization is intimately connected with the memory based certification. That is to say, one can not simply assign an arbitrary region of certification to every input point with the use of memory and expect to attain high certified accuracy.

- We added further discussions in the limitations section (Appendix E) on the dependence of the certification on the order of inferenced points. Moreover, we discussed reasons to why in all our experiments, our certification was order invariant.

- We complemented our results with $\ell_1$ certification in Appendix G. In particular, we compared against Yang et. al. (Randomized smoothing of all shapes and sizes) against our proposed memory-enhanced data dependent smooth classifier.

- We moved the runtime section to Appendix H to comply with the 9-pages constraints.

---

### Author Response · Authors · 2021-11-23
**General Comment**

We thank all the reviewers for their comments and feedback which has greatly improved our submission. We took our best shot in the rebuttal hoping that we have addressed all concerns with the provided extra experiments (that are now in the appendix). Moreover, we emphasized further (as suggested by reviewer viFi )that we certify the memory-enhanced data dependent smooth classifier across the paper to avoid confusion. Please let us know if there are any other questions or experiments that you think are essential towards improving the scores.

---

### Decision · Program_Chairs · 2022-01-20

**Decision:**

Reject

**Comment:**

The idea to adapt the noise variance in the certification of a base classifier sounds natural and interesting, but unfortunately fundamentally flawed, as correctly pointed out by Reviewer viFi (also acknowledged in the authors' response): the author's main algorithm does not lead to any theoretical certification while the empirical fix (based on memory), however successful in one's experiment, does not rule out the possibility of failure when future test samples flood in. Incidentally, I believe this fallacy may have also answered Reviewer Xsdx's question (why this has not been done before). I agree with Reviewer viFi that the writing of this work is a bit deceptive and will require significant change. In particular, one cannot wave hands at claims on certification: you need to formally prove the memory-based empirical fix will provably certify a region for what classifier and under what assumption. Therefore, the current draft cannot be accepted. Please consider rethinking about the idea and rewriting the paper according to the reviewers' comments.